# Multi-Marginal Schrödinger Bridge Matching

## Abstract

Understanding the continuous evolution of populations from discrete temporal snapshots is a critical research challenge, particularly in fields like developmental biology and systems medicine where longitudinal tracking of individual entities is often impossible. Such trajectory inference is vital for unraveling the mechanisms of dynamic processes. While Schrödinger Bridge (SB) offer a potent framework, their traditional application to pairwise time points can be insufficient for systems defined by multiple intermediate snapshots. This paper introduces Multi-Marginal Schrödinger Bridge Matching (MSBM), a novel algorithm specifically designed for the multi-marginal SB problem. MSBM extends iterative Markovian fitting (IMF) to effectively handle multiple marginal constraints. This technique ensures robust enforcement of all intermediate marginals while preserving the continuity of the learned global dynamics across the entire trajectory. Empirical validations on synthetic data and real-world single-cell RNA sequencing datasets demonstrate the competitive or superior performance of MSBM in capturing complex trajectories and respecting intermediate distributions, all with notable computational efficiency.

## 1 Introduction

Understanding the continuous evolution of populations from discrete temporal snapshots represents a significant challenge in various scientific disciplines, particularly in fields like developmental biology (Schiebinger et al., 2019; Bunne et al., 2023) and systems medicine (Manton et al., 2008) where tracking individual entities longitudinally is often unfeasible. The ability to infer trajectories from such snapshot data is crucial for elucidating the underlying mechanisms of dynamic processes. The Schrödinger Bridge (SB) problem, originally rooted in statistical mechanics (Schrödinger, 1931), has garnered substantial interest in machine learning as an entropy-regularized, continuous-time formulation of optimal transport (Léonard, 2013; Mikami, 2021). It seeks to identify the most probable evolutionary path between prescribed initial and terminal distributions, and has been successfully employed in generative modeling (Bortoli et al., 2021; Vargas et al., 2021; Chen et al., 2022a; Liu et al., 2023; Shi et al., 2024; Peluchetti, 2023; Liu et al., 2024b; Bortoli et al., 2024).

However, many real-world scenarios present observations or constraints at multiple time points, not just at the beginning and end of a process. For instance, in single-cell RNA sequencing (scRNA-seq) experiments, which are pivotal for studying complex biological processes like cell differentiation, cells are typically destroyed upon measurement (Macosko et al., 2015; Klein et al., 2015; Buenrostro et al., 2015). This destructive nature makes it impossible to track individual cells over time, thus necessitating the inference of developmental trajectories from population-level snapshots collected at several intermediate stages. Similarly, meteorological systems may have partial observations across various times (Moon et al., 2019; Chu et al., 2016). Such situations necessitate a multi-marginal generalization of the SB problem (mSBP), where the path measure must align with prescribed marginal distributions at multiple intermediate time points. While the traditional SB framework offers a powerful approach, its standard application to pairwise time points can prove insufficient for systems characterized by multiple intermediate snapshots. Although more specialized methods for mSBP have recently been developed (Koshizuka & Sato, 2022; Chen et al., 2023; Shen et al., 2024), the direct application of some multi-marginal approaches can lead to error accumulation if not carefully managed, particularly when learned controls are even slightly inaccurate. These challenges highlight the need for robust and scalable solutions for the mSBP that can effectively integrate information across all observed time points.

This paper introduces Multi-Marginal Schrödinger Bridge Matching (MSBM), a novel algorithm specifically developed to address the multi-marginal SB problem by building upon and extending the Iterative Markovian Fitting (IMF) algoritmh (Shi et al., 2024; Peluchetti, 2022). MSBM is designed to effectively manage multiple marginal constraints by constructing local SBs on each interval and seamlessly integrating them. This local construction strategy, underpinned by a shared global parametrization of control functions, ensures the robust enforcement of all intermediate marginal distributions while crucially preserving the continuity of the learned global dynamics across the entire trajectory. Empirical validations conducted on synthetic datasets as well as real-world single-cell RNA sequencing data demonstrate that MSBM achieves competitive or superior performance in capturing complex trajectories and accurately respecting intermediate distributions, all while exhibiting notable computational efficiency. Our work aims to provide a robust and scalable computational method for these multi-marginal settings, addressing the critical need for consistent and tractable dynamic inference when data is available as snapshots at multiple time points.

We summarize our contributions as follows:

- We extend the theoretical and algorithmic foundations of SBs, including the IMF iteration and optimal control perspectives, to the challenging multi-marginal setting.

- We introduce an efficient modeling approach for trajectory inference, that constructs and smoothly integrates local SBs across sub-intervals, inherently allows for parallelized training, leading to significant speed-ups.

- Through comprehensive experiments on both synthetic and real-world single-cell RNA sequencing data, we demonstrate that MSBM accurately models complex population dynamics and outperforms state-of-the-art methods in both trajectory fidelity and computational speed.

**Notation.** Let $\mathcal{P}_{[0,T]}$ denote the space of continuous functions taking values in $\mathbb{R}^d$ on the interval $[0,T]$. We use an uppercase letter $\mathbb{P} \in \mathcal{P}_{[0,T]}$ to represent a path measure. For a path measure $\mathbb{P} \in \mathcal{P}_{[0,T]}$, the marginal distribution at discrete time points $\mathcal{T} = \{t_0, \dots, t_k\}$, where $0 = t_0 < t_1 < \cdots < t_k = T$ is denoted by $\mathbb{P}_\mathcal{T} \in \mathcal{P}_\mathcal{T}$, where we define $\mathcal{P}_\mathcal{T}$ as the set of measures $\mathbb{P}$ over $\mathbb{R}^{d \times |\mathcal{T}|}$. Additionally, the conditional distribution of $\mathbb{P}$, given $\mathcal{T}$, is denoted by $\mathbb{P}_{|\mathcal{T}} \in \mathcal{P}_{[0,T]}$. Moreover, a path measure $\mathbb{P}$ can be defined as mixture. For any Borel measurable set $A \in \mathcal{B}(\Omega)$, $\mathbb{P}$ can be defined by $\mathbb{P}(A) = \int_{\mathbb{R}^{d \times |\mathcal{T}|}} \mathbb{P}_{|\mathcal{T}}(A|\mathbf{x}_\mathcal{T}) d\mathbb{P}_\mathcal{T}(\mathbf{x}_\mathcal{T})$, where $\mathbb{P} \in \mathcal{P}_{0,T}$ and $\mathbb{P} \in \mathcal{P}_\mathcal{T}$, and we use the shorthand $\mathbf{x}_\mathcal{T} := (\mathbf{x}_1, \cdots, \mathbf{x}_k)$ and $[0:k] := \{0, 1, \cdots, k\}$. The Kullback-Leibler (KL) divergence between two probability measures $\mu$ and $\nu$ on space $\mathcal{X}$ is defined as $D_{\mathrm{KL}}(\mu|\nu) = \int_\mathcal{X} \log \frac{d\mu}{d\nu}(\mathbf{X}) d\mu(\mathbf{X})$ when $\mu$ is absolutely continuous with respect to $\nu$ ($\mu \ll \nu$), and $D_{\mathrm{KL}}(\mu|\nu) = +\infty$ otherwise. We will often refer to probability measures on $\mathbb{R}^d$ and their Lebesgue densities interchangeably, under the standard assumption of absolute continuity. Finally, for a function $\mathcal{V} : [0,T] \times \mathbb{R}^d \to \mathbb{R}$, we define the gradient and Laplcaian operators with respect to $\mathbf{x} \in \mathbb{R}^d$ as $\nabla\mathcal{V}$ and $\Delta\mathcal{V}$, respectively, and its partial derivative with respect to time $t \in [0,T]$ as $\partial_t \mathcal{V}$.

## 2 Preliminaries

### 2.1 Schrödinger Bridge Matching (SBM)

The Schrödinger Bridge problem (SBP) (Schrödinger, 1931; Jamison, 1975) is a stochastic optimal transport problem that seeks the optimal transport plan for endpoint marginals $\rho_0$ and $\rho_T$. In this paper, we focus on the dynamical representation, where a reference distribution $\mathbb{Q} \in \mathcal{P}_{[0,T]}$ is induced by the SDEs:

$$d\mathbf{X}_t = f_t(\mathbf{X}_t)\,dt + \sigma\,d\mathbf{W}_t, \quad \mathbf{X}_0 \sim \rho_0, \tag{1}$$

where $f_t : \mathbb{R}^d \to \mathbb{R}^d$ is a drift, $\sigma \in \mathbb{R}$ is a diffusion, and $\mathbf{W}_t \in \mathbb{R}^d$ is a Brownian motion. With the base reference path measure $\mathbb{Q}$, the dynamic representation of the SB (Pavon & Wakolbinger, 1991; Pra, 1991) is:

$$\min_{\mathbb{P} \in \mathcal{P}_{[0,T]}} D_{\mathrm{KL}}(\mathbb{P}|\mathbb{Q}), \quad \text{subject to} \quad \mathbb{P}_0 \sim \rho_0, \quad \mathbb{P}_T \sim \rho_T. \tag{SBP}$$

Recent advancements in dynamical optimal transport (Peluchetti, 2023; Shi et al., 2024) have introduced a novel numerical methodology for solving SBP. This approach reframes SBP by decomposing its dynamical constraints into the time-evolving marginal distributions $\mathbb{P}_t$ for all $t \in [0, T]$ often characterized by drift function and the joint coupling $\mathbb{P}_{0,T}$ for a given end point marginal pairs $(\rho_0, \rho_T)$. This optimization relies on IMF (Shi et al., 2024), a technique that iteratively refines the path measure $\mathbb{P} \in \mathcal{P}_{[0,T]}$. IMF alternates between two projection called Markovian and Reciprocal projections to preserve the correct endpoint marginals $(\rho_0, \rho_T)$ throughout the optimization.

**Reciprocal Projection $\mathcal{R}$.** For a given reference measure $\mathbb{Q}$ from (1), and a path measure $\mathbb{P}$ with marginals specified at end points $\mathcal{T} = \{0, T\}$ the reciprocal projection is defined as:

$$\mathcal{R}(\mathbb{P}, \mathcal{T}) := \mathbb{P}_\mathcal{T} \mathbb{Q}_{|\mathcal{T}} = \mathbb{P}_{0,T} \mathbb{Q}_{|0,T}. \tag{2}$$

This projection constructs a new path measure by taking the endpoint coupling $\mathbb{P}_{0,T}$ from $\mathbb{P}$ and forming a mixture of bridge process using $\mathbb{Q}$ conditioned on these end points. Sampling from $\Pi := \mathcal{R}(\mathbb{P}, \mathcal{T})$ involves drawing end points samples $(\mathbf{X}_0, \mathbf{X}_T) \sim \mathbb{P}_{0,T}$ and then generating a path $\mathbf{X}_t^\mathcal{T}$ between them using conditional reference measure $\mathbb{Q}_{|0,T}$ which induced by following SDEs, for any $(\mathbf{x}_0, \mathbf{x}_T)$:

$$d\mathbf{X}_t^\mathcal{T} = \left[ f_t(\mathbf{X}_t^\mathcal{T}) + \sigma^2 \nabla \log \mathbb{Q}_{T|t}(\mathbf{x}_T | \mathbf{X}_t^\mathcal{T}) \right] dt + \sigma d\mathbf{W}_t, \quad \mathbf{X}_0^\mathcal{T} = \mathbf{x}_0, \tag{3}$$

If $\mathbb{Q}_{|0,T}$ has tractable bridge formulation, for example, when $\mathbb{Q}$ is chosen as a Brownian motion $i.e., d\mathbf{X}_t = \sigma d\mathbf{W}_t$, sampling the path at time $t$ given the endpoints can be performed as:

$$\mathbf{X}_t^\mathcal{T} \sim \mathcal{N}\left( (1 - \tfrac{t}{T})\mathbf{X}_0 + \tfrac{t}{T}\mathbf{X}_T, t(1 - \tfrac{t}{T})\sigma^2 \right), \quad \text{where } (\mathbf{X}_0, \mathbf{X}_T) \sim \mathbb{P}_{0,T}. \tag{4}$$

**Markov Projection $\mathcal{M}$.** Although the reciprocal projection $\mathcal{R}$ in (2) preserves end point marginals $(\rho_0, \rho_T)$, its sampling process in (4) requires both $(\mathbf{X}_0, \mathbf{X}_T)$, making it non-Markovian and thus ill-suited for generative modeling aimed at sampling from $\rho_T$ without knowing $\mathbf{X}_T$. The Markov projection $\mathcal{M}$ resolves this by projecting $\Pi := \mathcal{R}(\mathbb{P}, \mathcal{T})$ into a family of Markov process while ensuring $\mathbb{P}^\star = \Pi_t$ for all $t \in [0, T]$. Again, when $\mathbb{Q}$ is chosen as a Brownian motion $i.e., d\mathbf{X}_t = \sigma d\mathbf{W}_t$, the Markov projection of $\Pi$, $\mathbb{P}^\star = \mathcal{M}(\Pi, \mathcal{T})$, is induced by following SDEs:

$$d\mathbf{X}_t^\star = \sigma v^\star(t, \mathbf{X}_t^\star) dt + \sigma d\mathbf{W}_t, \quad \mathbf{X}_0^\star \sim \Pi_0, \tag{5}$$

$$\text{where} \quad v^\star(t, \mathbf{x}) = \tfrac{1}{T-t} \left( \mathbb{E}_{\mathbb{Q}_{T|t}} [\mathbf{X}_T | \mathbf{X}_t = \mathbf{x}] - \mathbf{x} \right). \tag{6}$$

Intuitively, the term $\mathbb{E}_{\mathbb{Q}_{T|t}} [\mathbf{X}_T | \mathbf{X}_t = \mathbf{x}]$ can be understood as a prediction of the target state $\mathbf{X}_t^\star$ (most probable pair for the given $\mathbf{X}_0^\star$). Recent matching based generative model (Lipman et al., 2023; Peluchetti, 2022) tackles the approximation $\mathbf{X}_T^\star := \mathbb{E}_{\mathbb{Q}_{T|t}} [\mathbf{X}_T | \mathbf{X}_t = \mathbf{x}] \approx v^\theta(t, \mathbf{x})$ by learning a parameterized drift function. This learned drift guides the evolution of $\mathbf{X}_t^\star$ such that its terminal state aligns with the optimal target, by regressing the drift against a target drift derived from samples of $(\mathbf{X}_0, \mathbf{X}_T)$ under the reference conditional bridge measure $\mathbb{Q}_{|0,T}$.

Building upon the projections $\mathcal{R}$ and $\mathcal{M}$, Schrödinger Bridge Matching (SBM) methods (Shi et al., 2024; Peluchetti, 2023) refines the path measure through an alternating iteraive procedure:

$$\mathbb{P}^{(2n+1)} := \mathcal{M}(\mathbb{P}^{(2n)}, \mathcal{T}), \ \mathbb{P}^{(2n+2)} := \mathcal{R}(\mathbb{P}^{(2n+1)}, \mathcal{T}). \tag{7}$$

Initialized with $\mathbb{P}^{(0)} = \mathbb{P}_\mathcal{T}^{(0)} \mathbb{Q}_{|0,T}$, utilizing $\mathbb{P}_\mathcal{T}^{(0)}$ is independent coupling of $\rho_0$ and $\rho_T$ along with the reference conditional bridge measure $\mathbb{Q}_{|\mathcal{T}}$. Please refer to (Shi et al., 2024; Peluchetti, 2023) for more details.

## 3 Multi-Marginal Iterative Markovian Fitting

Dynamic SB methods, as discussed in Section 2, have traditionally focused on problems defined by two endpoint marginal distributions, $(\rho_0, \rho_T)$. However, in real-world applications, particularly in fields like developmental biology (e.g., scRNA-seq studies of cellular differentiation), systems are often observed through

snapshots at multiple intermediate time points, not just at the beginning and end of a process. This prevalence of multi-stage data highlights a critical limitation of standard SB approaches. While the theoretical extension of SB methods to handle multiple marginals has been explored (Baradat & Léonard, 2020; Mohamed et al., 2021), the development of robust and scalable computational methods for these multi-marginal settings has lagged. Recently, methods with IPF-type objectives have been derived for multi-marginal cases (Chen et al., 2023; Shen et al., 2024). However, challenges persist in ensuring global dynamic consistency across all intervals, maintaining computational tractability as the number of marginals increases.

In this section, we extends the SBM framework−conventionally applied to problems with two endpoint marginals $(\rho_0, \rho_T)$−to handle cases involving $k+1$ multiple snapshots $(\rho_0, \rho_{t_1}, \cdots, \rho_T)$ on discrete time stamps $\mathcal{T} = \{t_0, t_1, \cdots, t_k\}$ where $0 = t_0 < t_1 < \cdots < t_k = T$[1]. Similar to SBP, the dynamic multi-marginal Schrödinger Bridge problem can be formally defined as (Chen et al., 2019) the entropy minimization problem:

$$\min_{\mathbb{P} \in \mathcal{P}_{[0,T]}} D_{\mathrm{KL}}(\mathbb{P}|\mathbb{Q}), \quad \text{subject to} \quad \mathbb{P}_t \sim \rho_t, \quad \forall t \in \mathcal{T}. \tag{mSBP}$$

To find a most probable path $\mathbb{P}^{\mathrm{mSBP}}$, the solution of mSBP under multiple constraints, we will generalize the principles of SBM in Section 2.1 to the multi-marginal cases in Section 3.1. The extension of dynamic SB optimality (Pavon & Wakolbinger, 1991; Léonard, 2013) and the associated stochastic optimal control problem (Pra, 1991) to multi-marginal settings is presented in Appendix A.

## 3.1 Multi-Marginal Projection Operators

To develop multi-marginal extension of SBM, we investigate how the IMF framework can be adapted to scenarios with multiple snapshots (*i.e.*, where the set of time points $\mathcal{T}$ has cardinality $|\mathcal{T}| > 2$). This adaptation necessitates extending the fundamental building blocks of SBM—specifically, the reciprocal projection $\mathcal{R}$ and the Markov projection $\mathcal{M}$—to handle multiple marginal constraints.

**Multi-Marginal Reciprocal Projection $\mathcal{R}^{\mathrm{mm}}$.** First, we state and prove a proposition that characterizes the reciprocal structure of conditional path measures. In particular, we focus on a mixture of bridges $\Pi = \Pi_{\mathcal{T}} \mathbb{Q}_{|\mathcal{T}} \in \mathbb{P}_{[0,T]}$ constrained by the marginals at multiple timestamps in $\mathcal{T}$.

**Proposition 1** (Reciprocal Property). *For any* $\mathbf{x}_{\mathcal{T}} := (\mathbf{x}_0, \mathbf{x}_{t_1}, \cdots, \mathbf{x}_T) \in \mathbb{R}^{d \times (k+1)}$ *and* $t \in [t_{i-1}, t_i)$, *the marginal distribution of* $\mathbb{Q}_{|\mathcal{T}}(\cdot|\mathbf{x}_{\mathcal{T}})$ *at* $t$ *satisfies:*

$$\mathbb{Q}_{|\mathcal{T}}(\mathbf{x}_t|\mathbf{x}_{\mathcal{T}}) = \mathbb{Q}_{|t_{i-1}, t_i}(\mathbf{x}_t|\mathbf{x}_{t_i}, \mathbf{x}_{t_{i-1}}). \tag{8}$$

*Therefore, for any* $\mathbb{P} \in \mathcal{P}_{[0,T]}$ *the reciprocal projection* $\mathcal{R}^{\mathrm{mm}}(\mathbb{P}, \mathcal{T})$ *admits the following factorization:*

$$\mathcal{R}^{\mathrm{mm}}(\mathbb{P}, \mathcal{T}) = \mathbb{P}_{\mathcal{T}} \mathbb{Q}_{|\mathcal{T}} = \mathbb{P}_{t_0, \cdots, t_k} \mathbb{Q}_{|t_0, \cdots, t_k} = \mathbb{P}_{t_0, \cdots, t_k} \prod_{i=1}^{k} \mathbb{Q}_{|t_{i-1}, t_i}, \quad \mathbb{P}\text{-}a.e. \tag{9}$$

A key implication of the reciprocal property, detailed in Proposition 1, is that a mixture of diffusion bridges constrained on $\mathcal{T}$ factorizes into independent segments over successive time intervals. This factorization simplifies the analysis and simulation of the overall path measure. Since each segment can then be treated as a standard conditional bridge process as in (3), closed-form sampling, such as in (4), can be applied independently in parallel to each subinterval $\{t_{i-1}, t_i\}_{i \in [1:k]}$. This tractability is essential for developing an efficient multi-marginal SBM algorithm.

**Multi-Marginal Markov Projection $\mathcal{M}^{\mathrm{mm}}$.** With the reciprocal property and factorization in (9), we show that the Markov projection on multi-marginal case can be constructed by similar fashion.

**Proposition 2** (Multi-Marginal Markovian Projection). *Let* $\Pi \in \mathcal{P}_{[0,T]}$ *admit factorzation in (9). The multi-marginal Markov projection of* $\Pi$, $\mathbb{P}^{\star} := \mathcal{M}^{\mathrm{mm}}(\Pi, \mathcal{T}) \in \mathcal{P}_{[0,T]}$, *is associated with the SDE:*

$$d\mathbf{X}_t^{\star} = [f_t(\mathbf{X}_t^{\star}) + \sigma v^{\star}(t, \mathbf{X}_t^{\star})] dt + \sigma d\mathbf{W}_t, \quad \mathbf{X}_0^{\star} \sim \Pi_0, \tag{10}$$

$$\text{where } v^{\star}(t, \mathbf{x}) = \sum_{i=1}^{k} \mathbf{1}_{[t_{i-1}, t_i)} \mathbb{E}_{\Pi_{t_i|t}} \left[ \nabla \log \mathbb{Q}_{t_i|t}(\mathbf{X}_{t_i}|\mathbf{X}_t)|\mathbf{X}_t = \mathbf{x} \right]. \tag{11}$$

---

[1]Our framework accommodates arbitrary time intervals between successive time stamps.

*Moreover, $v^\star$ satisfies the Fokker-Planck equation (FPE) (Risken & Risken, 1996):*

$$\partial_t \rho_t = -\nabla \cdot (v_t^\star(\mathbf{x})\rho_t(\mathbf{x})) + \tfrac{\sigma^2}{2}\Delta\rho_t(\mathbf{x}) = 0, \quad \rho_t = \Pi_t, \quad \forall t \in \mathcal{T}, \tag{12}$$

*where $p_t$ is marginal density of $\Pi_t$. In other words, $\mathbb{P}_t^\star = \Pi_t$ for all $t \in [0, T]$. d*

As established in Proposition 2, constructing a global diffusion process via (10) with the optimal control $v^\star$ (11)) yields a multi-marginal Markov projection $\mathbf{X}_{[0,T]}^\star$ that is continuous over the entire time interval $[0, T]$. The continuity arises because the local Markov projections, $\mathbf{X}_{[t_{i-1}, t_i]}^\star$, on each sub-interval are derived from factorized conditional bridge $\mathbb{Q}_{|t_{i-1}, t_i}$ in (9). These bridges are anchored by identical marginal distributions at there shared boundaries; for instance, both $\mathbf{X}_{[t_{i-1}, t_i]}^\star$ and $\mathbf{X}_{[t_i, t_{i+1}]}^\star$ is guaranteed to match the marginal distribution $\rho_{t_i}$ at time $t_i$. Consequently, these local diffusion processes connect seamlessly at adjacent timestamps, resulting in a continuous and well-defined path for $\mathbf{X}_{[0,T]}^\star$. The well-defined nature of the global path, in conjunction with the projections $\mathcal{R}^{\text{mm}}$ and $\mathcal{M}^{\text{mm}}$, is fundamental to successfully applying the SBM framework to the mSBP. Finally, the uniquness condition for standard SB (Shi et al., 2024, Proposition 5) can also be extended to multi-marginal case.

**Proposition 3** (Uniqueness). *Let $\mathbb{P}^\star$ be a Markov measure which is reciprocal class of $\mathbb{Q}$ satisfying $\mathbb{P}_t^\star = \rho_t$ for all $t \in \mathcal{T}$. Then, $\mathbb{P}^\star$ is unique solution $\mathbb{P}^{\text{mSBP}}$ of the mSBP.*

Building on the projection operators $\mathcal{R}^{\text{mm}}, \mathcal{M}^{\text{mm}}$ with the uniqueness result of Proposition 3, we can apply the iterative algorithm used in SBM algorithm (Shi et al., 2024, Algorithm 1) to the multi-marginal setting:

$$\mathbb{P}^{(2n+1)} := \mathcal{M}^{\text{mm}}(\mathbb{P}^{(2n)}, \mathcal{T}), \ \mathbb{P}^{(2n+2)} := \mathcal{R}^{\text{mm}}(\mathbb{P}^{(2n+1)}, \mathcal{T}), \quad |\mathcal{T}| > 2. \tag{13}$$

The convergence guarantees proved for the iteration apply equally well to the multi-marginal case.

**Proposition 4** (Convergence). *$\mathbb{P}^{(n)} = \mathbb{P}^{\text{mSBP}}$ of mSBP as $n \uparrow \infty$ with iterative procedure in (13).*

### 3.2 Practical Implementation.

In practice, at each iteration $n$ of (13) we approximate the optimal control $v^\star$ from (11) by a neural network $v_\theta$. By Girsanov theorem, $\theta$ are chosen to minimize the following training objective function:

$$\mathcal{L}(\theta, \mathcal{T}, \Pi_{\mathcal{T}}) = \int_0^T \mathbb{E}_{\Pi_{t,\mathcal{T}}}[||\sigma\nabla\log\mathbb{Q}_{\beta_{\mathcal{T}}(t)|t}(\mathbf{X}_{\beta_{\mathcal{T}}(t)}|\mathbf{X}_t) - v_\theta(t, \mathbf{X}_t)||^2 dt], \tag{14}$$

where $\beta_{\mathcal{T}}(t) = \min_u\{u > t | u \in \mathcal{T}\} \in [0, T]$ is the most recent time point in $\mathcal{T}$ after time $t$. With this notation, the SBM can be generalized to the case of multi-marginal constraints. For example, when $\mathcal{T} = \{0, T\}$ then (14) reduces to the objective function described in (Shi et al., 2024).

The learned Markov control $v_{\theta^\star}(t, \mathbf{x}_t)$ then ensures $\mathbb{P}_t^{\theta^\star} = \Pi_t$ for all $t \in [0, T]$. Moreover, prior SBM algorithms interleave forward and backward-time Markov projections to re-anchor the terminal distribution and prevent bias between $\mathbb{P}_T^{(n)}$ and $\Pi_T$ accumulate for each $n \in \mathbb{N}$. In the multi-marginal setting, we again build the backward-time Markov projection as in Proposition 2 by *gluing* the local bridge reversals, so that $\mathbb{P}^\star$ is governed by both SDEs (10) and the corresponding backward dynamics:

$$d\mathbf{Y}_t^\star = [-f_{T-t}(\mathbf{Y}_t^\star) + \sigma u^\star(t, \mathbf{Y}_t^\star)] dt + \sigma d\mathbf{W}_t, \quad \mathbf{Y}_0^\star \sim \Pi_T, \tag{15}$$

$$\text{where } u^\star(t, \mathbf{y}) = \sum_{i=1}^k \mathbf{1}_{(t_{i-1}, t_i]}(t)\mathbb{E}_{\Pi_{t|t_{i-1}}}\left[\nabla\log\mathbb{Q}_{t|t_{i-1}}(\mathbf{Y}_t|\mathbf{Y}_{t_{i-1}})|\mathbf{Y}_t = \mathbf{y}\right], \tag{16}$$

where the backward optimal control $u^\star$ in (16) can be approximated with neural network $u_\phi$ where $\phi$ is chosen to minimize the following training objective function with $\gamma_{\mathcal{T}}(t) = \max_u\{u < t | u \in \mathcal{T}\} \in [0, T]$:

$$\mathcal{L}(\phi, \mathcal{T}, \Pi_{\mathcal{T}}) = \int_0^T \mathbb{E}_{\Pi_{t,\mathcal{T}}}[||\sigma\nabla\log\mathbb{Q}_{t|\gamma_{\mathcal{T}}(t)}(\mathbf{Y}_t|\mathbf{Y}_{\gamma_{\mathcal{T}}(t)}) - u_\phi(t, \mathbf{Y}_t)||^2 dt]. \tag{17}$$

---

**Algorithm 1** Training of MSBM

1: **Input:** Snapshots $\{\rho_t\}_{t \in \mathcal{T}}$, bridge $\mathbb{Q}_{|\mathcal{T}}$, $N \in \mathbb{N}$
2: Let $\{\mathbb{P}_{\mathcal{T}_i}^{(0)}\}_{i \in [1:k]}$ joint coupling of $\{\rho_{t \in \mathcal{T}_i}\}_{i \in [1:k]}$.
3: **for** $n \in \{0, \dots, N-1\}$ **do**
4:     **for** $i \in \{1, \dots, k-1\}$ **do in parallel**
5:         Let $\Pi_{\mathcal{T}_i}^{(2n)} = \mathbb{P}_{\mathcal{T}_i}^{(2n)}$
6:         Estimate $\mathcal{L}(\phi, \mathcal{T}_i, \Pi_{\mathcal{T}_i}^{(2n)}, \mathbb{Q}_{|\mathcal{T}_i})$
7:         Estimate $\tilde{\mathcal{L}}(\phi) = \sum_{i=1}^{k} \mathcal{L}(\phi, \mathcal{T}_i, \Pi_{\mathcal{T}_i}^{(2n)}, \mathbb{Q}_{|\mathcal{T}_i})$
8:         $u_{\phi^\star} = \arg\min_\phi \sum_{i=1}^{k} \tilde{\mathcal{L}}(\phi)$
9:         Simulate local backward SBs $\{\mathbb{P}^{i,(2n+1)}\}_{i \in [1:k]}$
10:     **for** $i \in \{1, \dots, k-1\}$ **do in parallel**
11:         Let $\Pi_{\mathcal{T}_i}^{(2n+1)} = \mathbb{P}_{\mathcal{T}_i}^{(2n+1)}$
12:         Estimate $\mathcal{L}(\theta, \mathcal{T}_i, \Pi_{\mathcal{T}_i}^{(2n+1)}, \mathbb{Q}_{|\mathcal{T}_i})$
13:         Estimate $\tilde{\mathcal{L}}(\theta) = \sum_{i=1}^{k} \mathcal{L}(\theta, \mathcal{T}_i, \Pi_{\mathcal{T}_i}^{(2n+1)}, \mathbb{Q}_{|\mathcal{T}_i})$
14:         $v_{\theta^\star} = \arg\min_\theta \sum_{i=1}^{k} \mathcal{L}(\theta, \mathcal{T}_i, \Pi_{\mathcal{T}_i}^{(2n+1)})$
15:         Simulate local forward SBs $\{\mathbb{P}_{[t_{i-1}, t_i]}^{i,(2n+2)}\}$
16: **end for**
17: **Output:** $v_\theta^\star, u_\phi^\star$

---

**Algorithm 2** Simulation of MSBM (forward)

**Input:** Initial $\rho_0$, learned control $v_{\theta^\star}$
Sample $\mathbf{X}_0 \sim \rho_0$
Simulate forward SDE over $[0, T]$
$d\mathbf{X}_t^\star = [f_t + \sigma v_{\theta^\star}(t, \mathbf{X}_t^\star)]\, dt + \sigma d\mathbf{W}_t,$
**Output:** Trajectory $\mathbf{X}_{[0,T]}^\star$

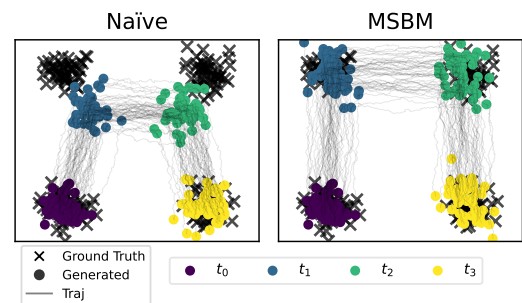

Figure 1: **(Left)** The naïve extension fails to model intermediate states due to the accumulation of errors. **(Right)** In contrast, MSBM successfully models the ground truth data.

## 4 Multi-Marginal Schrödinger Bridge Matching

A naïve extension of the standard SBM using, multi-marginal projections $\mathcal{R}^{\mathtt{mm}}$ and $\mathcal{M}^{\mathtt{mm}}$ in Sec 3, might encounters significant limitations which is not presented in the traditional two-endpoint setting. In such an extension, each iteration typically enforces marginal constraints only at the global endpoints $(\rho_0, \rho_T)$. The multi-marginal coupling $\Pi_{\mathcal{T}}^{(n)}$ at each iteration $n$ of (13) is then derived by propagating the projected dynamics in (10) or (15) solely from these end points $\rho_0$ or $\rho_T$, respectively.

**Error accumulation.** This approach leads to critical issues specific to the multi-marginal context. Firstly, if the learned controls, such as $v^\star$ (forward) or $u^\star$ (backward), are even slightly inaccurate, significant biases can arise between the inferred intermediate marginals $(\Pi_{t_1}^{(n)}, \cdots \Pi_{t_{k-1}}^{(n)})$ and the target marginals $(\rho_{t_1}, \cdots, \rho_{t_{k-1}})$. Secondly, these discrepancies tend to accumulate iteratively. This accumulation is exacerbated because, beyond an initialization $\Pi^{(0)} = \mathbb{P}_{\mathcal{T}}^{(0)} \mathbb{Q}_{|\mathcal{T}}$ with $\mathbb{P}_{\mathcal{T}}^{(0)}$, independent joint coupling of $\{\rho_t\}_{t \in \mathcal{T}}$, where the joint distribution might be informed by all prescribed data distributions, the subsequent self-refinement process for the dynamics often does not directly incorporate the intermediate data distributions $(\rho_{t_1}, \cdots, \rho_{t_{k-1}})$ into its training objective except $\rho_0$ and $\rho_T$. Without explicit targets for the intermediate marginals guiding each iteration, the inferred paths between $\rho_0$ and $\rho_T$ can *collapse* or drift away from the desired states. Consequently, precisely satisfying all intermediate constraints becomes increasingly challenging as iterations proceed as shown in Figure 1.

**Marginal-scaling Cost** Moreover, for each IMF iteration $n$, simulating Markov dynamics in (10) or (15) over the entire time interval $[0, T]$ to infer all the intermediate marginals $(\Pi_{t_1}^{(n)}, \cdots \Pi_{t_{k-1}}^{(n)})$. This creates a cost that grows with the number of marginals $K$ can be computational overhead for training.

To address this issues of error accumulation and computational overhead while multi-marginal optimality for (mSBP) satisfied, we propose a simple modifies that involves constructing local SBs on each interval $[t_{i-1}, t_i]$ and then seamlessly *gluing* them together. Instead of propagating dynamics from the global endpoints $\rho_0$ and $\rho_T$ alone, our approach first establishes local SBs for each segment. The results of each local SBs are then systematically integrated to satisfy all specified marginal distributions $\{\rho_t\}_{t \in \mathcal{T}}$ across the entire time interval $[0, T]$. This local construction strategy helps prevent the compounding of errors at intermediate time points while **parallel optimization** over each local interval is enabled. We first show theoretical basis that the gluing local SBs achieve the overall multi-marginal SB solution, $\mathbb{P}^{\mathtt{mSBP}}$, as below.

**Corollary 5** (Multi-Marginal Schrödinger Bridge). *Let us assume a sequence of controls $\{v^i, u^i\}_{i \in [1:k]}$, where each $v^i, u^i$ induced local SBs $\mathbb{P}^i$ of SBP over local interval $[t_{i-1}, t_i]$ with distributions $(\rho_{t_{i-1}}, \rho_{t_i})$ in a forward and backward direction, respectively. If $\lim_{t \uparrow t_i} v^i(t, \mathbf{x}) = v^{i+1}(t, \mathbf{x})$ and $\lim_{t \downarrow t_{i-1}} u^i(t, \mathbf{x}) = u^{i-1}(t, \mathbf{x})$ for all $i \in [1:k]$, then $\mathbb{P}^{mSBP}$ of mSBP induced by following SDEs:*

$$d\mathbf{X}_t^\star = [f_t(\mathbf{X}_t^\star) + \sigma v^\star(t, \mathbf{X}_t^\star)] \, dt + \sigma d\mathbf{W}_t, \quad \mathbf{X}_0^\star \sim \rho_0. \tag{18a}$$

$$d\mathbf{Y}_t^\star = [-f_{T-t}(\mathbf{Y}_t^\star) + \sigma u^\star(t, \mathbf{Y}_t^\star)] \, dt + \sigma d\mathbf{W}_t, \quad \mathbf{Y}_0^\star \sim \rho_T, \tag{18b}$$

$$where \quad v^\star(t, \mathbf{x}) = \sum_{i=1}^k \mathbf{1}_{[t_{i-1}, t_i)}(t) v^i(t, \mathbf{x}), \quad u^\star(t, \mathbf{x}) = \sum_{i=1}^k \mathbf{1}_{(t_{i-1}, t_i]}(t) u^i(t, \mathbf{x}). \tag{18c}$$

Building upon Corollary 5, we introduce our Multi-Marginal Schrödinger Bridge Matching (MSBM) method to solve the mSBP. A cornerstone of MSBM is divide the global mSBP into local SBPs while maintaining the continuity of the composite drift functions $v^\star$ and $u^\star$ in (18c) across adjacent intervals, which guarantees a globally continuous diffusion process inducing $\mathbb{P}^{mSBP}$. Furthermore, by explicitly constraining each local SBs, $\mathbb{P}^i$, on its corresponding marginals $(\rho_{t_{i-1}}, \rho_{t_i})$, MSBM is designed to mitigate the accumulation of bias at intermediate marginals, as shown in Figure 1.

A key challenge of the MSBM is rigorously satisfying the continuity conditions at the boundaries of local controls: $\lim_{t \uparrow t_i} v^i(t, \mathbf{x}) = v^{i+1}(t, \mathbf{x})$ and $\lim_{t \downarrow t_{i-1}} u^i(t, \mathbf{x}) = u^{i-1}(t, \mathbf{x})$ for all $i \in [1:k]$. If these conditions are not met, discontinuities or "kinks" can arise at the intermediate time steps. Such kinks would imply that the overall path measure $\mathbb{P}^\star \neq \mathcal{M}^{mm}(\mathbb{P}^\star, \mathcal{T})$. This would, in turn, hinder the optimality for mSBP, because, following Proposition 3, the desired continuous Markov process is a fixed point of both $\mathcal{R}^{mm}$ and Markov projections $\mathcal{M}^{mm}$ under multiple time points $\mathcal{T}$:

$$\mathbb{P}^\star = \mathcal{R}^{mm}(\mathbb{P}^\star, \mathcal{T}) = \mathcal{M}^{mm}(\mathbb{P}^\star, \mathcal{T}). \tag{19}$$

To construct local SBs such that the continuity requirements for forming a valid global solution are met, thereby preventing the aforementioned kinks and ensuring (19), our MSBM introduces a shared global parametrization $v_\theta, u_\phi$ for its respective local controls $\{v^i, u^i\}_{i \in [1:k]}$ for each sub-interval. The continuity of standard neural networks with respect to their input, hence seamlessly guarantees the continuous condition in 5 while each local controls can be updated in parallel manner with following aggregate objective function:

$$\tilde{\mathcal{L}}(\theta) = \sum_{i=1}^k \mathcal{L}(\theta, \mathcal{T}_i, \Pi_{\mathcal{T}_i}), \quad \tilde{\mathcal{L}}(\phi) = \sum_{i=1}^k \mathcal{L}(\phi, \mathcal{T}_i, \Pi_{\mathcal{T}_i}), \tag{20}$$

where $\mathcal{T}_i = \{t_{i-1}, t_i\}$ define sub-intervals with local coupling $\Pi_{\mathcal{T}_i}$ for end-points marginals in interval $[t_{i-1}, t_i]$ and $\mathcal{L}$ is defined in (14) and (17) for forward and backward direction, respectively.

The MSBM training procedure, summarized in Algorithm 1, adapts the standard IMF algorithm presented in (Shi et al., 2024, Algorithm 1). A key distinction in our MSBM approach is the parallel application of the IMF procedure to each local time interval, utilizing globally shared forward $v_\theta$ and backward $u_\phi$ across all local intervals. This parallel processing across sub-intervals contributes to a significant reduction in overall training time.

## 5 Related Work

The solution of SBP often utilize Iterative Proportional Fitting (IPF) (Kullback, 1968), with modern adaptations learning SDE drifts for two-marginal settings (Bortoli et al., 2021; Vargas et al., 2021; Chen et al., 2022a; Deng et al., 2024). A distinct iterative approach, IMF, as featured in (Shi et al., 2024; Peluchetti, 2023), offers improved stability by alternating projections onto different classes of path measures. Moreover, emerging research also explores non-iterative algorithm (De Bortoli et al., 2024; Peluchetti, 2025). These methodologies primarily concentrate on the SB problem itself, iteratively refining path measures or directly computing the bridge measure. Moreover, the SB algorithm is studied under the assumption that the optimal coupling is given (Liu et al., 2023; Somnath et al., 2023). While recent studies have extended foundational SB ideas to the multi-marginal setting of mSBP, research in this area remains relatively limited.

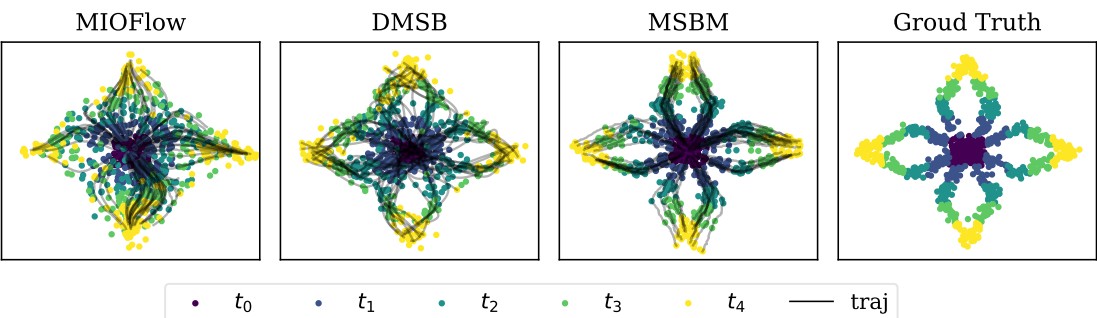

Figure 2: Comparison of generated population dynamics using MIOFlow, DMSB and MSBM on a 2-dim petal dataset. All trajectories are generated by simulating the dynamics from $\rho_{t_0}$.

In multi-marginal setting, (Chen et al., 2023) extends the problem to phase space to encourage smoother trajectories and introduces a novel training methodology inspired by the Bregman iteration (Bregman, 1967) to handle multiple marginal constraints. Relatedly, (Shen et al., 2024) presented an approach that, similar to our work, segments the problem across intervals; they learn piecewise SBs and use likelihood-based training to iteratively refine a global reference dynamic. While these methods are often IPF-based or focus on specific reference refinement strategies, our MSBM extends the previous IMF-type algorithm into multi-marginal setting and effectively handles multiple constraints. We demonstrate that our MSBM framework offers substantial gains in training efficiency. This enhanced efficiency is primarily attributed to its direct multi-marginal formulation that adeptly manages multiple constraints, thereby circumventing the computationally intensive iterative refinements common in IPF-based methods

Paralleling these SB-centric developments, other significant lines of work model dynamic trajectories by directly learning potential functions or velocity fields, often drawing from optimal transport or continuous normalizing flows. For instance, (Koshizuka & Sato, 2022; Liu et al., 2022; 2024b;a) extend SBs to incorporate potentials or mean-field interactions, connecting to stochastic optimal control and earlier mean-field game frameworks (Ruthotto et al., 2020; Lin et al., 2021). The broader field of trajectory inference from snapshot data, crucial for applications like scRNA-seq, has seen methods like (Tong et al., 2020) using CNFs with dynamic OT, and (Huguet et al., 2022) employing Neural ODEs on learned data manifolds. More recently, (Neklyudov et al., 2023a;b) offer variational objectives to learn dynamics from marginal samples.

## 6 Experiments

We empirically demonstrate the effectiveness of our MSBM. Specifically, our goal is to infer a dynamic model from datasets composed of samples from marginal distributions $\rho_t$ observed at discrete time points. We evaluate MSBM on both synthetic datasets and real-world single-cell RNA sequencing datasets, including human embryonic stem cells (hESC) (Chu et al., 2016), embryoid body (EB) (Moon et al., 2019) CITE-seq (CITE) and Multiome (MULTI) (Burkhardt et al., 2022). To ensure consistency and fair comparison, our experiments follow the respective experimental setups established by baseline methods. In particular, for the petal dataset, we adopt the experimental setup from DMSB (Chen et al., 2023), and for the hESC dataset, we follow SBIRR (Shen et al., 2024). For the EB dataset, we perform evaluations on both 5-dim and 100-dim PCA-reduced data; here, we follow the 100-dim experimental setup of DMSB and the 5-dim setup from NLSB (Koshizuka & Sato, 2022). For the CITE and MULTI datasets, we follow the setup in (Tong et al., 2023; 2024). Accordingly, we follow evaluation protocols and utilize evaluation metrics consistent with previous studies, including the Sliced-Wasserstein Distance (SWD)(Bonneel et al., 2015), Maximum Mean Discrepancy (MMD)(Gretton et al., 2012), as well as the 1-Wasserstein ($\mathcal{W}_1$) and 2-Wasserstein ($\mathcal{W}_2$) distances. All experimental results reported are averaged mean value over three independent runs with different random seeds. We leave further experimental details in Appendix C.

## 6.1 Synthetic Data

**Petal** The petal dataset (Huguet et al., 2022) serves as a simple yet complex challenge because it mimics the natural dynamics seen in processes such as cellular differentiation, which include phenomena like bifurcations and merges. We compare our MSBM with MIOFlow (Huguet et al., 2022) and DMSB (Chen et al., 2023) in Figure 3. As shown in Figure 2, we observe that MSBM exhibits the most accurate and clearly defined trajectory, closely resembling the ground truth. Furthermore, Figure 3 demonstrates the evaluation results for the trajectories through $\mathcal{W}_2$ and MMD distances, highlighting that MSBM consistently outperforms MIOFlow and DMSB.

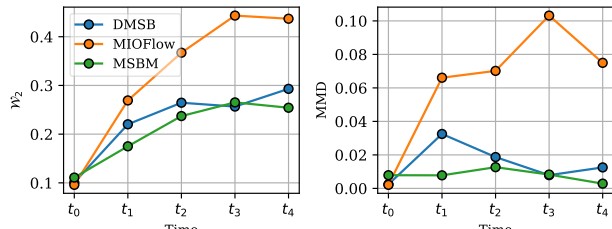

Figure 3: Evaluation results of $\mathcal{W}_2$ and MMD.

## 6.2 Single-cell Sequencing Data

We evaluated our MSBM on real-world single-cell RNA sequencing data from two sources: **1**) human embryonic stem cells (hESCs) (Chu et al., 2016) undergoing differentiation into definitive endoderm over a 4-day period, measured at 6 distinct time points ($t_0$:0 hours, $t_1$:12 hours, $t_2$:24 hours, $t_3$:36 hours, $t_4$:72 hours, and $t_5$:96 hours); **2**) embryoid body (EB) cells (Moon et al., 2019) differentiating into mesoderm, endoderm, neuroectoderm, and neural crest over 27 days, with samples collected at 5 time windows ($t_0$:0-3 days, $t_1$:6-9 days, $t_2$:12-15 days, $t_3$:18-21 days, and $t_4$:24-27 days). Following the experimental setup of baselines, we preprocessed these datasets using the pipeline outlined in (Tong et al., 2020), and the collected cells were projected into a lower-dimensional space using principal component analysis (PCA).

**hESC** To follow the experimental setup from SBIRR (Shen et al., 2024), we reduced the data to the first five principal components and excluded the final time point $t_6$ from our dataset, resulting in three training time points $\mathcal{T} = \{t_0, t_2, t_4\}$ and two intermediate test points $\mathcal{T}_{\texttt{test}} = \{t_1, t_3\}$. Our objective was to train the dynamics based on the available marginals at the training points in $\mathcal{T}$ and interpolate the intermediate test marginals at $\mathcal{T}_{\texttt{test}}$, which were not observed during training. Table 1 demonstrates that our proposed MSBM method performs competitively, achieving lower $\mathcal{W}_2$ distances.

Table 1: Performance on the 5-dim PCA of hESC dataset. $\mathcal{W}_2$ is compute between test $\rho_{t_i}$ and generated $\hat{\rho}_{t_i}$ by simulating the dynamics from test $\rho_{t_0}$. Best results are highlighted.

| Methods | $\mathcal{W}_2 \downarrow$ | | Runtime |
| | $t_1$ | $t_3$ | hours |
| --- | --- | --- | --- |
| DMSB[†] | 1.10 | 1.51 | 15.54 |
| SBIRR[†] | **1.08** | 1.33 | 0.36 (0.38)* |
| **MSBM** (Ours) | 1.09 | **1.30** | **0.09** |

† result from (Shen et al., 2024).

**Embryoid Body** We validate our MSBM on both 5-dim and 100-dim PCA spaces. First, for the 5-dim experiment, we adopt the experimental setup from (Koshizuka & Sato, 2022). Given 5 observation points $\mathcal{T} = \{t_0, t_1, t_2, t_3, t_4\}$, we divide the data train/test splits $\rho_{\mathcal{T}}^{\texttt{tr}}/\rho_{\mathcal{T}}^{\texttt{te}}$, with the goal of predicting population dynamics from $\rho_{t_0}^{\texttt{tr}}$. We train the dynamics based on $\rho_{\mathcal{T}}^{\texttt{tr}}$ and evaluate the $\mathcal{W}_1$ distance between $\rho_{t_i}^{\texttt{te}}$ and the generated $\hat{\rho}_{t_i}$ from previous test snapshot $\rho_{t_{i-1}}^{\texttt{te}}$. In Table 2, we find that MSBM outperforms baseline SB methods.

For the 100-dim experiment, we borrow the experimental setup from DMSB, where the goal is predict population dynamics given that observations are available for all time points $\mathcal{T}$ (denoted as Full in Table 3), or when one of the snapshot is left out (denoted as $t_i$ in Table 3, where snapshot $\rho_{t_i}^{\texttt{tr}}$ at $t_i$ is excluded during training). The high performance in this task represent the robustness of the model to accurately predict population dynamics. In Table 3, MSBM consistently yields performance improvements. Moreover, as shown in Figure 4, the trajectories and generated marginal distributions $\hat{\rho}_{\mathcal{T}}$

Table 2: Performance on the 5-dim PCA of EB dataset. $\mathcal{W}_1$ is computed between test $\rho_{t_i}^{\texttt{te}}$ and generated $\hat{\rho}_{t_i}$ by simulating the dynamics from previous test $\rho_{t_{i-1}}^{\texttt{te}}$. Best results are highlighted.

| Methods | $\mathcal{W}_1 \downarrow$ | | | | |
| | $t_1$ | $t_2$ | $t_3$ | $t_4$ | Mean |
| --- | --- | --- | --- | --- | --- |
| NLSB[†] (Koshizuka & Sato, 2022) | 0.68 | 0.84 | 0.81 | 0.79 | 0.78 |
| IPF (GP)[†] (Vargas et al., 2021) | 0.70 | 1.04 | 0.94 | 0.98 | 0.92 |
| IPF (NN)[†] (Bortoli et al., 2021) | 0.74 | 0.89 | 0.84 | 0.83 | 0.82 |
| SB-FBSDE[†] (Chen et al., 2022a) | **0.56** | 0.80 | 1.00 | 1.00 | 0.84 |
| MSBM (Ours) | 0.64 | **0.73** | **0.72** | **0.73** | **0.71** |

† result from (Koshizuka & Sato, 2022).

Table 3: Performance on the 100-dim PCA of EB dataset. MMD and SWD are computed between test $\rho_{t_i}^{\mathtt{te}}$ and generated $\hat{\rho}_{t_i}$ by simulating the dynamics from test $\rho_{t_0}^{\mathtt{te}}$. Best results are highlighted.

| Methods | MMD ↓ | | | | SWD ↓ | | | |
|---|---|---|---|---|---|---|---|---|
| | Full | $t_1$ | $t_2$ | $t_3$ | Full | $t_1$ | $t_2$ | $t_3$ |
| NLSB[†] | 0.66 | 0.38 | 0.37 | 0.37 | 0.54 | 0.55 | 0.54 | 0.55 |
| DMSB[†] | 0.03 | **0.04** | **0.04** | **0.04** | 0.16 | 0.20 | 0.19 | **0.18** |
| **MSBM** | **0.02** | **0.04** | **0.04** | 0.05 | **0.11** | **0.18** | **0.17** | 0.19 |

† result from (Chen et al., 2023).

Figure 4: Comparison of generated population dynamics using DMSB and MSBM on a 100-dim PCA of EB dataset. The plot displays the first two principal components as the x and y axes, respectively.

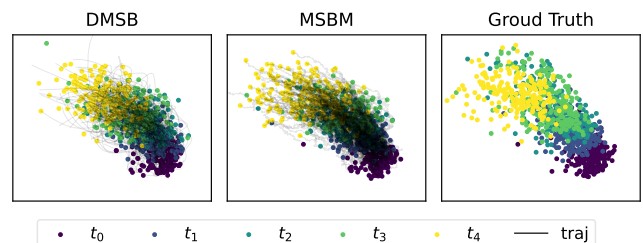

Table 4: Performance on the 5-dim and 100-dim PCA of CITE/MULTI dataset. $\mathcal{W}_1$ are computed between test $\rho_{t_i}^{\mathtt{te}}$ and generated $\hat{\rho}_{t_i}$ by simulating the dynamics from test $\rho_{t_{i-1}}^{\mathtt{te}}$. Best results are highlighted.

| Model | CITE | | MULTI | | Runtime |
|---|---|---|---|---|---|
| | 5dim | 100dim | 5dim | 100dim | hours |
| OT-CFM[†] | 0.88±0.06 | **45.39±0.42** | 0.94±0.05 | 54.81±5.86 | - |
| WLF-SB[†] | **0.80±0.02** | 46.13±0.08 | 0.95±0.21 | 55.07±5.50 | 0.51 |
| DMSB | 0.82±0.04 | 50.24±0.99 | 1.03±0.12 | 58.51±5.11 | 2.16 |
| **MSBM** | 0.87±0.05 | 45.93±0.37 | **0.91±0.22** | **54.75±6.21** | **0.08** |

† result from (Kapusniak et al., 2024).

in PCA space further justifies the numerical result and highlights the variety and quality of the samples produced by MSBM.

**CITE-seq and Multiome**   Finally, we compare with the flow-matching variants such as OT-CFM (Tong et al., 2024) and the OT-based method WLF-SB (Neklyudov et al., 2023b). Here, we perform a leave-one-timepoint-out evaluation. In particular, for each $t_i$, we train on all marginals except $t_i$ and evaluate using the $\mathcal{W}_1$ distance between the predicted marginal $\hat{\rho}_{t_i}$ and the held-out marginal $\rho_{t_i}^{\mathtt{te}}$. We validate MSBM in 5- and 100-dimensional PCA spaces. Table 4 shows that MSBM achieves performance comparable on the CITE dataset and the best results on the MULTI dataset in both low- and high-dimensional settings.

**Computational Efficiency**   Moreover, we also measure end-to-end training time on the CITE-seq 100-dim using the same hardware for every method (a single NVIDIA RTX 3090 GPU). Although MSBM is not the state-of-the-art in every configuration, it still clearly outperforms all baselines in training speed. In particular, MSBM achieves a total runtime more than 27× faster than DMSB. Note that OT-CFM is excluded because an implementation details for this specific task is unavailable, which prevents a fair comparison.

This enhanced computational efficiency primarily originates from core algorithmic differences. DMSB employs an IPF-type objective with Bregman Iteration (Bregman, 1967). In contrast, MSBM directly optimizes controls using an IMF-type objective, which not only eliminates the need to store intermediate states but also facilitates parallel computation across sub-intervals. This approach substantially promotes faster convergence of the algorithm.

## 7   Conclusion and Limitation

This paper revisits previously established frameworks for the SBP, extending them to the mSBP. Specifically, we introduce a computationally efficient framework for mSBP, termed MSBM, which builds upon existing SBM methods (Shi et al., 2024; Peluchetti, 2023). MSBM is tailored for various trajectory inference problems where snapshots of data are available at multi-marginal time steps. Through the successful adaptation of the IMF algorithm to this multi-marginal setting, our approach significantly accelerates training processes while ensuring accurate dynamic modeling when compared to existing methods.

Despite these advantages, the performance degradation of MSBM is more pronounced than that of DMSB when a time point is omitted in Table 3. This may occur because the including velocity term could better accommodate unknown trajectory. Furthermore, the current MSBM framework is restricted to the case involving snapshot data samples, highlighting a need for enhancements to address problems with continuous potentials (Chen et al., 2022b; Liu et al., 2024b), such mean-field games (Liu et al., 2022; 2024a).

### Broader Impact Statement

This paper presents work aimed at advancing the field of machine learning. Our research may have various societal consequences. However, we do not believe any of these require specific emphasis here.

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

# A   Control Theoretic Multi-Marginal Schrödinger Bridge Formulation

In this section, we examine the structure of the solution and the reduction of the Schrödinger Bridge problem in the multi-marginal setting. Since the case of two marginals has been thoroughly studied in prior works (Léonard, 2013; Chen et al., 2021), we aim to directly extend these results to the multi-marginal scenario.

In standard SBP, the optimality condition can be characterized in terms of Doob's $h$-transform (Jamison, 1975; Chen et al., 2021), with two potential functions $(\overrightarrow{\Phi}, \overleftarrow{\Phi})$ that satisfy the forward and backward time harmonic equations.

**Theorem A.1** (Dynamics SB optimality (Pavon & Wakolbinger, 1991)). *Let $(\overrightarrow{\Phi}, \overleftarrow{\Phi})$ be the solutions to the following PDEs:*

$$\partial_t \overrightarrow{\Psi} + \nabla \overrightarrow{\Phi}^\top f + \tfrac{1}{2}\sigma^2 \Delta \overrightarrow{\Psi} = 0, \quad \partial_t \overleftarrow{\Phi} + \nabla \cdot (\overleftarrow{\Phi} f) - \tfrac{1}{2}\sigma^2 \Delta \overleftarrow{\Phi} = 0, \tag{21}$$

$$\rho_0(\mathbf{x}) = \overrightarrow{\Phi}(0, \mathbf{x})\overleftarrow{\Phi}(0, \mathbf{x}), \rho_T(\mathbf{x}) = \overrightarrow{\Psi}(T, \mathbf{x})\overleftarrow{\Phi}(T, \mathbf{x}). \tag{22}$$

*Then, the solution $\mathbb{P}^{SBP}$ of SBP is induced by following forward-backward SDEs:*

$$d\overrightarrow{\mathbf{X}}_t = \left[ f_t + \sigma^2 \nabla_{\mathbf{x}} \overrightarrow{\Phi}(t, \overrightarrow{\mathbf{X}}_t) \right] dt + \sigma d\mathbf{W}_t, \quad \overrightarrow{\mathbf{X}}_0 \sim \rho_0, \tag{23a}$$

$$d\overleftarrow{\mathbf{X}}_t = \left[ -f_{T-t} + \sigma^2 \nabla_{\mathbf{x}} \overleftarrow{\Phi}(T-t, \overleftarrow{\mathbf{X}}_t) \right] dt + \sigma d\mathbf{W}_t, \quad \overleftarrow{\mathbf{X}}_0 \sim \rho_T. \tag{23b}$$

Note that the forward-backward stochastic process in (23a) and (23b) satisfying $\rho_t^{(23a)} = \rho_t^{(23a)} = \rho_t$ for all $t \in [0, T]$ and $\rho_t$ obeys a factorization $\rho_t(\mathbf{x}) = \overrightarrow{\Phi}(t, \mathbf{x})\overleftarrow{\Phi}(t, \mathbf{x})$. To solve the SB, one needs to solve the associated PDEs to estimate $(\overrightarrow{\Phi}, \overleftarrow{\Phi})$. However, due to the high-dimensional nature of the problem, directly solving the PDEs is challenging (Han et al., 2018). Previous works (Bortoli et al., 2021; Chen et al., 2022a; Liu et al., 2022) has addressed this issue by formulating an IPF type algorithm (Kullback, 1968), where half-bridge optimization is iteratively repeated for the two boundary conditions:

$$\mathbb{P}^{(n+1)} := \underset{\mathbb{P} \in \mathcal{P}_{[0,T]}, \mathbb{P}_T = \rho_T}{\arg\min} D_{\mathrm{KL}}(\mathbb{P}|\mathbb{P}^{(n)}), \ \mathbb{P}^{(n+2)} := \underset{\mathbb{P} \in \mathcal{P}_{[0,T]}, \mathbb{P}_0 = \rho_0}{\arg\min} D_{\mathrm{KL}}(\mathbb{P}|\mathbb{P}^{(n+1)}). \tag{24}$$

with initialization $\mathbb{P}^{(0)} := \mathbb{Q}$. Please refer to (Bortoli et al., 2021) for more details.

Now, we extend the SB optimality in A.1 to multi-marginal settings by defining appropriate potentials. Let $\mathbb{Q} \in \mathcal{P}_{[0,T]}$ be a path measure induced by following Itô SDEs $d\mathbf{X}_t = \sigma d\mathbf{W}_t$. We assume that a collection of $k + 1$ marginal distributions $\rho_{\mathcal{T}} := (\rho_0, \rho_{t_1}, \cdots, \rho_{t_k})$ is specified at timestamps $\mathcal{T} = \{t_0, t_1, \cdots t_k\}$ where $= 0 = t_0 < t_1 \cdots < t_k = T$. We want to explore the most likely evolution between multiple marginals $\rho_{\mathcal{T}}$ which is the solution of multi-marginal SBP:

$$\min_{\mathbb{P} \in \mathcal{P}(\Omega)} D_{\mathrm{KL}}(\mathbb{P}|\mathbb{Q}), \quad \text{subject to} \quad \mathbb{P}_t \sim \rho_t, \quad \forall t \in \mathcal{T}. \tag{mSBP}$$

We first consider the static problem of mSBP. Consider the conditioning of the process on $\mathbf{X}_0 = \mathbf{x}_0, \mathbf{X}_{t_1} = \mathbf{x}_{t_1}, \cdots, \mathbf{X}_T = \mathbf{x}_{t_k}$ such that

$$\mathbb{P}_{|\mathcal{T}} = \mathbb{P}\left[\cdot | \mathbf{X}_0 = \mathbf{x}_0, \mathbf{X}_{t_1} = \mathbf{x}_{t_1}, \cdots, \mathbf{X}_T = \mathbf{x}_T\right] \tag{25}$$

$$\mathbb{Q}_{|\mathcal{T}} = \mathbb{Q}\left[\cdot | \mathbf{X}_0 = \mathbf{x}_0, \mathbf{X}_{t_1} = \mathbf{x}_{t_1}, \cdots, \mathbf{X}_T = \mathbf{x}_T\right]. \tag{26}$$

These conditioned laws can be interpreted as the disintegration of path measure $\mathbb{P}, \mathbb{Q} \in \mathcal{P}_{[0,T]}$ with respect to multiple time points. Given the multi-marginal joint distributions $\mathcal{T}, \mathbb{P}_{\mathcal{T}}, \mathbb{Q}_{\mathcal{T}} \in \mathbb{P}_{\mathcal{T}}$ associated with $\mathbb{P}$ and $\mathbb{Q}$ at timestamps $\mathcal{T}$, respectively, the relative entropy admits the following decomposition:

$$D_{\mathrm{KL}}(\mathbb{P}|\mathbb{Q}) = \underbrace{D_{\mathrm{KL}}(\mathbb{P}_{\mathcal{T}}|\mathbb{Q}_{\mathcal{T}})}_{\text{static}} + \underbrace{\int_{\mathbb{R}^{d \times |\mathcal{T}|}} D_{\mathrm{KL}}(\mathbb{P}_{|\mathcal{T}}|\mathbb{Q}_{|\mathcal{T}})d\mathbb{P}_{\mathcal{T}}}_{\text{dynamic}}. \tag{27}$$

Since $\mathbb{P}_{\mathcal{T}}$ and $\mathbb{P}_{|\mathcal{T}}$ can be chosen arbitrarily, we may set $\mathbb{P}_{|\mathcal{T}} = \mathbb{Q}_{|\mathcal{T}}$ so that the `dynamic` component of the decomposition in (27) vanishes. Under this choice, `mSBP` reduces to the `static` problem:

$$\min_{\mathbb{P}_{\mathcal{T}} \in \Pi_{\mathcal{T}}} D_{\mathrm{KL}}(\mathbb{P}_{\mathcal{T}} | \mathbb{Q}_{\mathcal{T}}), \tag{smSBP}$$

$$\text{where} \quad \Pi_{\mathcal{T}} = \{\mathbb{P}_{\mathcal{T}} \in \mathbb{R}^{d \times |\mathcal{T}|} \big| \int_{d \times (|\mathcal{T}|-1)} \mathbb{P}_{\mathcal{T}}(\mathbf{x}_{\mathcal{T}^{(-i)}}) d\mathbf{x}_{\mathcal{T}^{(-i)}} = \rho_{t_i}, \forall i \in [0:k]\}, \tag{28}$$

where $\mathcal{T}^{(-i)} = \{0, \cdots, t_{i-1}, t_{i+1}, \cdots, T\}$ denotes the set of indices excluding $t_i$. In other words, this formulation seeks a multi-marginal coupling $\mathbb{P}_{\mathcal{T}}$ that matches the prescribed marginal $\rho_{\mathcal{T}}$, while minimizing its relative entropy with respect to the reference joint law $\mathbb{Q}_{\mathcal{T}}$. Therefore, once the optimal coupling $\Pi_{\mathcal{T}}^{\star} = \mathbb{P}_{\mathcal{T}}^{\star}$ solving `smSBP` is obtained, it directly induces the solution to `mSBP` via

$$\mathbb{P}^{\star}(\cdot) = \int_{\mathbb{R}^{d \times |\mathcal{T}|}} \mathbb{Q}_{|\mathcal{T}}(\cdot) d\mathbb{P}_{\mathcal{T}}^{\star}, \tag{29}$$

meaning that $\mathbb{P}^{\star}$ constructed in this way satisfies the original `mSBP`.

Now, we want to derive the multi-marginal Schrödinger potential for `mSBP`. To do so, let us consider the Lagrangian formulation $\mathcal{L}$ of `mSBP` which is given by:

$$\mathcal{L}(\mathbb{P}_{\mathcal{T}}, \lambda_{\mathcal{T}}) = D_{\mathrm{KL}}(\mathbb{P}_{\mathcal{T}} | \mathbb{Q}_{\mathcal{T}}) + \sum_{i=0}^{|\mathcal{T}|} \int_{\mathbb{R}^d} \lambda_{t_i}(\mathbf{x}_{t_i}) \left[ \int_{\mathbb{R}^{d \times (\mathcal{T}-1)}} \mathbb{P}_{\mathcal{T}}(\mathbf{x}_{\mathcal{T}}) d\mathbf{x}_{\mathcal{T}^{(-i)}} - \rho_{t_i}(\mathbf{x}_{t_i}) \right] d\mathbf{x}_{t_i}, \tag{30}$$

where $\lambda_{\mathcal{T}} = (\lambda_0, \lambda_{t_1}, \cdots, \lambda_T)$ is the Lagrange multipliers where each $\lambda_{t_i}$ enforcing the marginal constraints at $t_i \in \mathcal{T}$. By setting the first variation of $\mathcal{L}(\mathbb{P}_{\mathcal{T}}, \lambda_{\mathcal{T}})$ to zero, we obtain the optimality condition for the minimizer $\mathbb{P}_{\mathcal{T}}^{\star}$. Let $\delta\mathbb{P}_{\mathcal{T}}(\mathbf{x}_{\mathcal{T}})$ be a perturbation around $\mathbb{P}_{\mathcal{T}}$. Consider the measure

$$\mathbb{P}_{\mathcal{T}}^{\epsilon} = \mathbb{P}_{\mathcal{T}} + \epsilon \delta\mathbb{P}_{\mathcal{T}}, \tag{31}$$

and by plugging $\mathbb{P}_{\mathcal{T}}^{\epsilon}$ in (31) into Lagrangian (30) and compute the first variation of each term. Then, for the first term of RHS in (30) we get:

$$\frac{d}{d\epsilon}\Big|_{\epsilon=0} D_{\mathrm{KL}}(\mathbb{P}_{\mathcal{T}}^{\epsilon} | \mathbb{Q}_{\mathcal{T}}) = \frac{d}{d\epsilon}\Big|_{\epsilon=0} \int \log \frac{d(\mathbb{P}_{\mathcal{T}} + \epsilon\delta\mathbb{P}_{\mathcal{T}})}{d\mathbb{Q}_{\mathcal{T}}} d(\mathbb{P}_{\mathcal{T}} + \epsilon\delta\mathbb{P}_{\mathcal{T}}) \tag{32}$$

$$= \int d\delta\mathbb{P}_{\mathcal{T}} \left( \log \frac{d\mathbb{P}_{\mathcal{T}}}{d\mathbb{Q}_{\mathcal{T}}} + 1 \right). \tag{33}$$

Moreover, for the second term of RHS in (30) we get:

$$\frac{d}{d\epsilon}\Big|_{\epsilon=0} \sum_{i=0}^{|\mathcal{T}|} \int_{\mathbb{R}^d} \lambda_{t_i}(\mathbf{x}_{t_i}) \left[ \int_{\mathbb{R}^{d \times (\mathcal{T}-1)}} (\mathbb{P}_{\mathcal{T}} + \epsilon\delta\mathbb{P}_{\mathcal{T}})(\mathbf{x}_{\mathcal{T}}) d\mathbf{x}_{\mathcal{T}^{(-i)}} - \rho_{t_i}(\mathbf{x}_{t_i}) \right] d\mathbf{x}_{t_i} \tag{34}$$

$$= \sum_{i=0}^{|\mathcal{T}|} \int_{\mathbb{R}^d} \lambda_{t_i}(\mathbf{x}_{t_i}) \left[ \int_{\mathbb{R}^{d \times (\mathcal{T}-1)}} \delta\mathbb{P}_{\mathcal{T}}(\mathbf{x}_{\mathcal{T}}) d\mathbf{x}_{\mathcal{T}^{(-i)}} \right] d\mathbf{x}_{t_i} \tag{35}$$

$$= \int_{\mathbb{R}^{d \times |\mathcal{T}|}} \left[ \sum_{i=0}^{|\mathcal{T}|} \lambda_{t_i}(\mathbf{x}_{t_i}) \right] d\delta\mathbb{P}_{\mathcal{T}}(\mathbf{x}_{\mathcal{T}}). \tag{36}$$

Hence, we get the total first variation of $\mathcal{L}(\mathbb{P}_{\mathcal{T}}^{\epsilon}, \lambda_{\mathcal{T}})$ as:

$$\delta\mathcal{L} = \int d\delta\mathbb{P}_{\mathcal{T}} \left( \log \frac{d\mathbb{P}_{\mathcal{T}}}{d\mathbb{Q}_{\mathcal{T}}} + 1 + \left[ \sum_{i=0}^{|\mathcal{T}|} \lambda_{t_i}(\mathbf{x}_{t_i}) \right] \right). \tag{37}$$

For the optimality, $\delta\mathcal{L}$ to be vanished for all admissible $\delta\mathbb{P}_{\mathcal{T}}$, hence we get:

$$\log \frac{d\mathbb{P}_{\mathcal{T}}}{d\mathbb{Q}_{\mathcal{T}}} + 1 + \left[ \sum_{i=0}^{|\mathcal{T}|} \lambda_{t_i}(\mathbf{x}_{t_i}) \right] = 0. \tag{38}$$

Therefore, we obtain the optimality condition:

$$\mathbb{P}^\star_\mathcal{T}(\mathbf{x}_\mathcal{T}) = \mathbb{Q}_\mathcal{T}(\mathbf{x}_\mathcal{T}) \exp\left(-1 - \left[\sum_{i=0}^{|\mathcal{T}|} \lambda_{t_i}(\mathbf{x}_{t_i})\right]\right) \tag{39}$$

$$\overset{(i)}{=} \prod_{i=1}^{|\mathcal{T}|} \mathbb{Q}_{|t_{i-1},t_i}(\mathbf{x}_{t_{i-1}}, \mathbf{x}_{t_i}) \exp\left(\log \rho_0(\mathbf{x}_0) - 1 - \left[\sum_{i=0}^{|\mathcal{T}|} \lambda_{t_i}(\mathbf{x}_{t_i})\right]\right) \tag{40}$$

$$\overset{(ii)}{=} \Psi_0(\mathbf{x}_0) \prod_{i=1}^{|\mathcal{T}|} \left[\mathbb{Q}_{|t_{i-1},t_i}(\mathbf{x}_{t_{i-1}}, \mathbf{x}_{t_i}) \Psi_{t_i}(\mathbf{x}_{t_i})\right] \tag{41}$$

where $(i)$ follows from $\mathbb{Q}_\mathcal{T}(\mathbf{x}_\mathcal{T}) = \rho_0(\mathbf{x}_0) \prod_{i=1}^k \mathbb{Q}_{|t_{i-1},t_i}(\mathbf{x}_{t_{i-1}}, \mathbf{x}_{t_i})$ and $(ii)$ follows from defining

$$\exp\left(\log \rho_0(\mathbf{x}_0) - 1 - \left[\sum_{i=0}^{|\mathcal{T}|} \lambda_{t_i}(\mathbf{x}_{t_i})\right]\right) = \prod_{i=0}^{|\mathcal{T}|} \Psi_{t_i}(\mathbf{x}_{t_i}), \tag{42a}$$

$$\text{where } \Psi_0 = \exp\left(\log \rho_0(\mathbf{x}_0) - 1 - \lambda_0\right), \text{ and } \Psi_{t_i} = \exp(-\lambda_{t_i}), \; \forall i > 0. \tag{42b}$$

Observe that since $\mathbb{P}^\star_\mathcal{T} = \Pi^\star_\mathcal{T}$ is the optimal coupling, it follows from equation (29) that the path measure factorizes as $d\mathbb{P}^\star = d\mathbb{P}^\star_\mathcal{T} \cdot \mathbb{Q}^\star_{|\mathcal{T}}$. Combining this with the result from equation (39), we obtain the Radon–Nikodym derivative:

$$\frac{d\mathbb{P}^\star}{d\mathbb{Q}} = \frac{d\mathbb{P}^\star_\mathcal{T}}{d\mathbb{Q}_\mathcal{T}} \cdot \frac{d\mathbb{P}^\star_{|\mathcal{T}}}{d\mathbb{Q}_{|\mathcal{T}}} = \frac{d\mathbb{P}^\star_\mathcal{T}}{d\mathbb{Q}_\mathcal{T}} = \prod_{i=0}^{|\mathcal{T}|} \Psi_{t_i}(\mathbf{x}_{t_i}). \tag{43}$$

Moreover, due to the structure of the construction and by applying results such as those in (Baradat & Léonard, 2020, Theorem 2.10), the resulting measure $\mathbb{P}^\star$ is a Markov process.

Hence, from (41), we deduce that the optimal coupling $\mathbb{P}^\star_\mathcal{T}$ factorized into a functions of $\mathbf{x}_{t_i}$. Moreover, since the conditional measure $\mathbb{Q}_{|t_{i-1},t_i}$ is reciprocal process, it satisfies time symmetry (Léonard et al., 2014) i.e., $\overrightarrow{\mathbb{Q}}_{|t_{i-1},t_i} = \overleftarrow{\mathbb{Q}}_{|t_i,t_{i-1}}$. Consequently, for any increasing time sequence $\overrightarrow{\mathcal{T}} = \{t_0, \cdots, t_k\}$ and its reversed counterpart $\overleftarrow{\mathcal{T}} = \{t_k, \cdots, t_0\}$, for a given $\mathbf{x}_\mathcal{T} \in \mathbb{R}^{d \times |\mathcal{T}|}$, it holds

$$\overrightarrow{\mathbb{Q}}_{|\overrightarrow{\mathcal{T}}} = \prod_{i=1}^{k} \overrightarrow{\mathbb{Q}}_{|t_{i-1},t_i} = \prod_{i=k}^{1} \overleftarrow{\mathbb{Q}}_{|t_i,t_{i-1}} = \overleftarrow{\mathbb{Q}}_{|\overleftarrow{\mathcal{T}}}. \tag{44}$$

We now define the multi-marginal Schrödinger potentials in both the forward and backward directions.

**Theorem A.2** (Multi-Marginal Schrödinger Potentials). *For a finite set of timestamps $\mathcal{T}$ and with corresponding potentials $\{\Psi_i\}_{i=0}^{|\mathcal{T}|}$ defined in (42), let us define a pair of potentials $(\overrightarrow{\Psi}, \overleftarrow{\Psi})$:*

$$\overrightarrow{\Psi}(t, \mathbf{x}) = \mathbb{E}_\mathbb{Q}\left[\prod_{j \geq \tau(t)}^{|\mathcal{T}|} \Psi_j(\mathbf{X}_{t_j}) | \mathbf{X}_t = \mathbf{x}\right], \quad \overleftarrow{\Psi}(t, \mathbf{x}) = \mathbb{E}_\mathbb{Q}\left[\prod_{j < \tau(t)}^{0} \Psi_j(\mathbf{X}_{t_j}) | \mathbf{X}_t = \mathbf{x}\right], \tag{45}$$

*where $\tau(t) = \min_u\{u \geq t | t \in \mathcal{T}\}$. Then, for any $t \in [t_{j-1}, t_j]$, $(\overrightarrow{\Psi}, \overleftarrow{\Psi})$ satisfy following PDEs:*

$$\partial_t \overrightarrow{\Psi} + \tfrac{1}{2}\sigma^2 \Delta \overrightarrow{\Psi} = 0, \quad \partial_t \overleftarrow{\Psi} - \tfrac{1}{2}\sigma^2 \Delta \overleftarrow{\Psi} = 0, \tag{46}$$

$$\rho_{t_j}(\mathbf{x}) = \overrightarrow{\Psi}(t_j, \mathbf{x})\overleftarrow{\Psi}(t_j, \mathbf{x}) = \lim_{t \uparrow t_j} \overrightarrow{\Psi}(t, \mathbf{x})\overleftarrow{\Psi}(t, \mathbf{x}), \tag{47}$$

$$\rho_{t_{j-1}}(\mathbf{x}) = \overrightarrow{\Psi}(t_{j-1}, \mathbf{x})\overleftarrow{\Psi}(t_{j-1}, \mathbf{x}) = \lim_{t \downarrow t_{j-1}} \overrightarrow{\Psi}(t, \mathbf{x})\overleftarrow{\Psi}(t, \mathbf{x}). \tag{48}$$

*Moreover, the solution $\mathbb{P}^{mSBP}$ of mSBP is induced by following forward-backward SDEs:*

$$d\overrightarrow{\mathbf{X}}_t = \sigma^2 \nabla_\mathbf{x} \overrightarrow{\Psi}(t, \overrightarrow{\mathbf{X}}_t)dt + \sigma d\mathbf{W}_t, \quad \overrightarrow{\mathbf{X}}_0 \sim \rho_0, \tag{49a}$$

$$d\overleftarrow{\mathbf{X}}_t = \sigma^2 \nabla_\mathbf{x} \overleftarrow{\Psi}(t, \overleftarrow{\mathbf{X}}_t)dt + \sigma d\mathbf{W}_t, \quad \overleftarrow{\mathbf{X}}_0 \sim \rho_T. \tag{49b}$$

*Proof.* We begin by proving the boundary conditions in (47–48). Without loss of generality, we consider the case where $t \in [t_{j-1}, t_j)$. By invoking the factorization in (43), we obtain:

$$\rho(t, \mathbf{x}) = \mathbb{E}_{\mathbb{P}^\star} \left[ \delta(\mathbf{X}_t = \mathbf{x}) \right] = \int_\Omega \delta(\mathbf{X}_t = \mathbf{x}) d\mathbb{P}^\star \tag{50}$$

$$= \int_\Omega \delta(\mathbf{X}_t = \mathbf{x}) e^{-1 - \sum_{i=0}^{|\mathcal{T}|} \lambda_{t_i}(\mathbf{x}_{t_i})} d\mathbb{Q} \tag{51}$$

$$= \int_{\mathbb{R}^{d \times |\mathcal{T}|}} e^{-1 - \sum_{i=0}^{|\mathcal{T}|} \lambda_{t_i}(\mathbf{x}_{t_i})} d\mathbb{Q}_{t,\mathcal{T}} \tag{52}$$

To evaluate the marginal at time $t$, we consider an integral over the path space $\Omega$, slicing the trajectory into segments for $t$. Due to the Markovian $\mathbb{Q}$, as consequence of (Léonard et al., 2014, Proposition 1.4) and time symmetry of bridge kernel (43), we can express the joint distribution $\mathbb{Q}_{t,\mathcal{T}}(\mathbf{x}_t, \mathbf{x}_\mathcal{T})$:

$$\mathbb{Q}_{t,\mathcal{T}} = \rho_0(\mathbf{x}_0) \prod_{i=1}^{j-1} \overrightarrow{\mathbb{Q}}_{|t_{i-1}, t_i} \prod_{i=j}^{|\mathcal{T}|} \overleftarrow{\mathbb{Q}}_{|t_{i-1}, t_i} \left[ \overrightarrow{\mathbb{Q}}_{|t_{j-1}, t} \overleftarrow{\mathbb{Q}}_{|t, t_j} \right]. \tag{53}$$

Hence, substituting $\mathbb{Q}_{t,\mathcal{T}}$ in (53) into (52), we get[2]:

$$\rho(t, \mathbf{x}) = \int_{\mathbb{R}^{d \times |\mathcal{T}|}} e^{\log \rho_0(\mathbf{x}_0) - 1 - \sum_{i=0}^{|\mathcal{T}|} \lambda_{t_i}(\mathbf{x}_{t_i})} \prod_{i=1}^{j-1} \mathbb{Q}_{|t_{i-1}, t_i} \prod_{i=j}^{|\mathcal{T}|} \mathbb{Q}_{|t_{i-1}, t_i} \left[ \mathbb{Q}_{|t_{j-1}, t} \mathbb{Q}_{|t, t_j} \right] d\mathbf{x}_\mathcal{T} \tag{54}$$

$$= \int_{\mathbb{R}^{d \times |\mathcal{T}|}} \prod_{i=0}^{|\mathcal{T}|} \Psi_{t_i}(\mathbf{x}_{t_i}) \prod_{i=1}^{j-1} \mathbb{Q}_{|t_{i-1}, t_i} \prod_{i=j}^{|\mathcal{T}|} \mathbb{Q}_{|t_{i-1}, t_i} \left[ \mathbb{Q}_{|t_{j-1}, t} \mathbb{Q}_{|t, t_j} \right] d\mathbf{x}_\mathcal{T} \tag{55}$$

$$= \overrightarrow{\Psi}(t, \mathbf{x}) \overleftarrow{\Psi}(t, \mathbf{x}). \tag{56}$$

Moreover, limiting procedures in (47-48) follows directly from the definition of (45): as time approaches the boundary $t_i$, the potential component $\Psi_i$, previously contained within the conditional expectation $\overrightarrow{\Psi}$ (or $\overleftarrow{\Psi}$), transitions smoothly into the complementary potential $\overleftarrow{\Psi}$ (or $\overrightarrow{\Psi}$) associated with the opposite time direction. This ensures the resulting density $\rho_t(\cdot) \in C^{1,2}([0, T], \mathbb{R}^d)$.

Now, as a consequence of (Mohamed et al., 2021, Theorem 4.19) for a unique minimizer of `mSBP` (if it exists), we get

$$D_{\mathrm{KL}}(\mathbb{P}^\star | \mathbb{Q}) = \frac{1}{2} \mathbb{E}_{\mathbb{P}^\star} \left[ \int_0^T \left\| \sigma \nabla \log \overrightarrow{\Psi}(t, \overrightarrow{\mathbf{X}}_t) \right\|^2 dt \right]. \tag{57}$$

In other words, by applying the Girsanov theorem (Øksendal, 2010, Theorem 8.6.5), it implies that the forward SDEs in (49a) induce the solution $\mathbb{P}^\star$ to `mSBP`. Consequently, the associated marginal density $\rho$ satisfies the Fokker-Planck equation:

$$\partial_t \rho - \nabla \cdot \left( \rho(\sigma^2 \nabla \overrightarrow{\Psi}) \right) - \frac{1}{2} \sigma^2 \Delta \rho = 0, \quad \rho_t(\mathbf{x}) = \overrightarrow{\Psi}_t(\mathbf{x}) \overleftarrow{\Psi}_t(\mathbf{x}), \quad \forall t \in \mathcal{T}. \tag{58}$$

Furthermore, by applying the Feynman-Kac formula (Baldi, 2017, Theorem 10.5), we obtain the PDE satisfied by $\overrightarrow{\Psi}$ as characterized in (Park et al., 2025, Appendix B.4):

$$\partial_t \overrightarrow{\Psi} + \tfrac{1}{2} \sigma^2 \Delta \overrightarrow{\Psi} = 0. \tag{59}$$

Combining equations (58–59) and using the factorization $\rho_t(\mathbf{x}) = \overrightarrow{\Psi}_t(\mathbf{x}) \overleftarrow{\Psi}_t(\mathbf{x})$, we deduce, following the derivation (Liu et al., 2022, Appendix A.4.1), that $\overleftarrow{\Psi}$ satisfies the backward PDE:

$$\partial_t \overleftarrow{\Psi} - \tfrac{1}{2} \sigma^2 \Delta \overleftarrow{\Psi} = 0. \tag{60}$$

Invoking Nelson's relation (Nelson, 1967), we recover the backward dynamics described in (49b), which likewise induce the optimal path measure $\mathbb{P}^\star$. This concludes the proof. □

---

[2]We will omit the arrow above the character, as it can be inferred from the subscript.

**Remark A.3.** *The IPF-type iteration (24) is expected to remain a valuable tool for approximating the multi-marginal potentials presented in Theorem A.2. Nevertheless, the challenge of enforcing boundary conditions for intermediate states, as specified in equations (47) and (48), might necessitate more advanced optimization strategies comparable to those developed in MSBM.*

Finally, combining (57-60), we obtain the stochastic optimal control formulation of mSBP:

$$\min_{\alpha \in \mathbb{A}} \mathcal{J}(\alpha) = \mathbb{E}_{\mathbb{Q}^\alpha} \left[ \int_0^T \frac{1}{2} \|\alpha_t\|^2 \, dt \right], \tag{61}$$

$$\text{subject to } d\mathbf{X}_t^\alpha = \sigma\alpha_t dt + \sigma d\mathbf{W}_t, \quad \mathbf{X}_t^\alpha \sim \rho_t, \forall t \in \mathcal{T}, \tag{62}$$

where $\mathbb{A}$ denotes the family of finite-energy controls adapted to the filtration generated by $\mathbf{W}_t$, satisfying $\mathbb{E}_{\mathbb{Q}^\alpha} \left[ \int_0^T \|\alpha_t\|^2, dt \right] < \infty$. Here, $\mathbb{Q}^\alpha$ is the path measure induced associated with (62).

Since the cost functional (61) can also be derived using Girsanov's theorem with reference measure $\mathbb{Q}$ associated to the uncontrolled SDE $d\mathbf{X}_t = \sigma d\mathbf{W}_t$, we conclude that the expression in (57) provides the optimal cost, *i.e.*, $\min \mathcal{J} = \frac{1}{2}\mathbb{E}_{\mathbb{P}^\star} \left[ \int_0^T \left\| \sigma\nabla \log \overrightarrow{\Psi}(t, \overrightarrow{\mathbf{X}}_t) \right\|^2 dt \right]$. Consequently, we identify the optimal control for the problem (61) as the Markovian feedback control $\alpha^\star := \sigma\nabla \log \overrightarrow{\Psi}$. Furthermore, this optimal control ensures that the marginal constraints in (62) are satisfied, as implied by (58).

## B  Proofs and Derivations

### B.1  Proof of Proposition 1

**Proposition 1** (Reciprocal Property). *For any $\mathbf{x}_\mathcal{T} := (\mathbf{x}_0, \mathbf{x}_{t_1}, \cdots, \mathbf{x}_T) \in \mathbb{R}^{d \times (k+1)}$ and $t \in [t_{i-1}, t_i)$, the marginal distribution of $\mathbb{Q}_{|\mathcal{T}}(\cdot|\mathbf{x}_\mathcal{T})$ at $t$ satisfies:*

$$\mathbb{Q}_{|\mathcal{T}}(\mathbf{x}_t|\mathbf{x}_\mathcal{T}) = \mathbb{Q}_{|t_{i-1},t_i}(\mathbf{x}_t|\mathbf{x}_{t_i}, \mathbf{x}_{t_{i-1}}). \tag{8}$$

*Therefore, for any $\mathbb{P} \in \mathcal{P}_{[0,T]}$ the reciprocal projection $\mathcal{R}^{mm}(\mathbb{P}, \mathcal{T})$ admits the following factorization:*

$$\mathcal{R}^{mm}(\mathbb{P}, \mathcal{T}) = \mathbb{P}_\mathcal{T}\mathbb{Q}_{|\mathcal{T}} = \mathbb{P}_{t_0,\cdots,t_k}\mathbb{Q}_{|t_0,\cdots,t_k} = \mathbb{P}_{t_0,\cdots,t_k}\prod_{i=1}^k \mathbb{Q}_{|t_{i-1},t_i}, \quad \mathbb{P}\text{-}a.e. \tag{9}$$

*Proof.* Let us consider a Markov measure $\mathbb{Q}$. Then following factorzation holds for $\mathbb{Q}$ (Baradat & Léonard, 2020, Definition 2.2) for any events $A_i \in \sigma(\mathbf{X}_{[t_{i-1},t_i]})$ for all $i \in [1:k]$:

$$\mathbb{Q}\left(\cap_{i=1}^k A_i \mid \mathbf{X}_\mathcal{T}\right) = \mathbb{Q}_{|0,t_1}(A_1 \mid \mathbf{X}_0, \mathbf{X}_{t_1})\mathbb{Q}_{|t_1,t_2}(A_2 \mid \mathbf{X}_{t_1}, \mathbf{X}_{t_2}) \cdots \mathbb{Q}_{|t_{k-1},T}(A_k \mid \mathbf{X}_{t_{k-1}}, \mathbf{X}_T). \tag{63}$$

Without loss of generality, consider $t \in [t_{i-1}, t_i)$. To isolate the conditional distribution of $\mathbf{X}_t$ given endpoints $\mathbf{X}_{t_{i-1}}$ and $\mathbf{X}_{t_i}$, choose events as follows:

$$A_j = \begin{cases} \Omega, & j \neq i, \\ \{\mathbf{X}_t \in B\}, & B \in \sigma(\mathbf{X}_t), \quad j = i. \end{cases} \tag{64}$$

Then, substituting (64) into the factorization in (63), we obtain:

$$\mathbb{Q}_{|\mathcal{T}}(\mathbf{X}_t \in B \mid \mathbf{X}_\mathcal{T}) = \mathbb{Q}_{|t_{i-1},t_i}(\mathbf{X}_t \in B \mid \mathbf{X}_{t_{i-1}}, \mathbf{X}_{t_i}). \tag{65}$$

Since $B \in \sigma(\mathbf{X}_t)$ was chosen arbitrarily, this implies that:

$$\mathbb{Q}_{|\mathcal{T}}(\mathbf{X}_t \mid \mathbf{X}_\mathcal{T}) = \mathbb{Q}_{|t_{i-1},t_i}(\mathbf{X}_t \mid \mathbf{X}_{t_{i-1}}, \mathbf{X}_{t_i}), \quad t \in [t_{i-1}, t_i). \tag{66}$$

Now, by disintegration of the path measure, we have

$$\mathbb{Q}(\cdot) = \int_{\mathbb{R}^{d \times |\mathcal{T}|}} \mathbb{Q}_{|\mathcal{T}}(\cdot|\mathbf{X}_{\mathcal{T}})d\mathbb{Q}_{\mathcal{T}}(\mathbf{X}_{\mathcal{T}}) = \int_{\mathbb{R}^{d \times |\mathcal{T}|}} \prod_{i=1}^{k} \mathbb{Q}_{|t_{i-1},t_i}(\cdot|\mathbf{X}_{t_{i-1},t_i})d\mathbb{Q}_{\mathcal{T}}(\mathbf{X}_{\mathcal{T}}), \mathbb{Q}\text{-a.e.} \tag{67}$$

It implies that for any $\mathbb{P}_{\mathcal{T}} \in \mathcal{P}_{\mathcal{T}}$, we get $\mathbb{P}_{\mathcal{T}}\mathbb{Q}_{|\mathcal{T}} = \mathbb{P}_{\mathcal{T}} \prod_{i=1}^{k} \mathbb{Q}_{|t_{i-1},t_i}$.

$\square$

## B.2 Proof of Proposition 2

**Proposition 2** (Multi-Marginal Markovian Projection). *Let $\Pi \in \mathcal{P}_{[0,T]}$ admit factorzation in (9). The multi-marginal Markov projection of $\Pi$, $\mathbb{P}^{\star} := \mathcal{M}^{mm}(\Pi, \mathcal{T}) \in \mathcal{P}_{[0,T]}$, is associated with the SDE:*

$$d\mathbf{X}_t^{\star} = [f_t(\mathbf{X}_t^{\star}) + \sigma v^{\star}(t, \mathbf{X}_t^{\star})] dt + \sigma d\mathbf{W}_t, \quad \mathbf{X}_0^{\star} \sim \Pi_0, \tag{10}$$

$$\text{where } v^{\star}(t, \mathbf{x}) = \sum_{i=1}^{k} \mathbf{1}_{[t_{i-1}, t_i)} \mathbb{E}_{\Pi_{t_i|t}} \left[ \nabla \log \mathbb{Q}_{t_i|t}(\mathbf{X}_{t_i}|\mathbf{X}_t)|\mathbf{X}_t = \mathbf{x} \right]. \tag{11}$$

*Moreover, $v^{\star}$ satisfies the Fokker-Planck equation (FPE) (Risken & Risken, 1996):*

$$\partial_t \rho_t = -\nabla \cdot (v_t^{\star}(\mathbf{x})\rho_t(\mathbf{x})) + \tfrac{\sigma^2}{2}\Delta\rho_t(\mathbf{x}) = 0, \quad \rho_t = \Pi_t, \quad \forall t \in \mathcal{T}, \tag{12}$$

*where $p_t$ is marginal density of $\Pi_t$. In other words, $\mathbb{P}_t^{\star} = \Pi_t$ for all $t \in [0, T]$. d*

*Proof.* Let $\mathcal{T}_i = \{t_{i-1}, t_i\}$ denote the set of two consecutive boundary time points, $\mathcal{T}_{<i} = \{0, \dots, t_{i-2}\}$ and $\mathcal{T}_{>i} = \{t_{i+1}, \dots, T\}$ represent the set of all time points preceding and following the interval $\mathcal{T}_i$, respectively. Then, for $t \in [t_{i-1}, t_i)$, we have:

$$\partial_t \rho_t = \partial_t \int \mathbb{Q}_{|\mathcal{T}}(\mathbf{x}_t)d\Pi_{\mathcal{T}}(\mathbf{x}_{\mathcal{T}}) \tag{68}$$

$$= \partial_t \int \frac{\mathbb{Q}_{t,\mathcal{T}}(\mathbf{x}_t, \mathbf{x}_{\mathcal{T}})}{\mathbb{Q}_{\mathcal{T}}(\mathbf{x}_{\mathcal{T}})}d\Pi_{\mathcal{T}}(\mathbf{x}_{\mathcal{T}}) \tag{69}$$

$$= \partial_t \int \frac{\mathbb{Q}_{t,\mathcal{T}_{<i},\mathcal{T}_{>i}|\mathcal{T}_i}(\mathbf{x}_t, \mathbf{x}_{\mathcal{T}_{<i}}, \mathbf{x}_{\mathcal{T}_{>i}}|\mathbf{x}_{\mathcal{T}_i})\mathbb{Q}_{\mathcal{T}_i}(\mathbf{x}_{\mathcal{T}_i})}{\mathbb{Q}_{\mathcal{T}}(\mathbf{x}_{\mathcal{T}})}d\Pi_{\mathcal{T}}(\mathbf{x}_{\mathcal{T}}) \tag{70}$$

$$\stackrel{(i)}{=} \partial_t \int \frac{\mathbb{Q}_{\mathcal{T}_{<i},\mathcal{T}_{>i}|\mathcal{T}_i}(\mathbf{x}_{\mathcal{T}_{<i}}, \mathbf{x}_{\mathcal{T}_{>i}}|\mathbf{x}_{\mathcal{T}_i})\mathbb{Q}_{t|\mathcal{T}_i}(\mathbf{x}_t|\mathbf{x}_{\mathcal{T}_i})\mathbb{Q}_{\mathcal{T}_i}(\mathbf{x}_{\mathcal{T}_i})}{\mathbb{Q}_{\mathcal{T}}(\mathbf{x}_{\mathcal{T}})}d\Pi_{\mathcal{T}}(\mathbf{x}_{\mathcal{T}}) \tag{71}$$

$$= \partial_t \int \frac{\mathbb{Q}_{\mathcal{T}}(\mathbf{x}_{\mathcal{T}})\mathbb{Q}_{t|\mathcal{T}_i}(\mathbf{x}_t|\mathbf{x}_{\mathcal{T}_i})}{\mathbb{Q}_{\mathcal{T}}(\mathbf{x}_{\mathcal{T}})}d\Pi_{\mathcal{T}}(\mathbf{x}_{\mathcal{T}}) \tag{72}$$

$$= \partial_t \int \mathbb{Q}_{t|\mathcal{T}_i}(\mathbf{x}_t|\mathbf{x}_{\mathcal{T}_i})d\Pi_{\mathcal{T}}(\mathbf{x}_{\mathcal{T}}) \tag{73}$$

$$= \int \partial_t \mathbb{Q}_{t|\mathcal{T}_i}(\mathbf{x}_t|\mathbf{x}_{\mathcal{T}_i})d\Pi_{\mathcal{T}}(\mathbf{x}_{\mathcal{T}}) \tag{74}$$

$$= \int \left[ -\nabla \cdot (v_{t|\mathcal{T}_i}(\mathbf{x}_t|\mathbf{x}_{\mathcal{T}_i})\rho_{t|\mathcal{T}_i}(\mathbf{x}_t|\mathbf{x}_{\mathcal{T}_i})) + \tfrac{1}{2}\sigma^2\Delta\rho_{t|\mathcal{T}_i}(\mathbf{x}_t|\mathbf{x}_{\mathcal{T}_i}) \right] d\Pi_{\mathcal{T}}(\mathbf{x}_{\mathcal{T}}) \tag{75}$$

$$= -\nabla \cdot \int v_{t|\mathcal{T}_i}(\mathbf{x}_t|\mathbf{x}_{\mathcal{T}_i})\rho_{t|\mathcal{T}_i}(\mathbf{x}_t|\mathbf{x}_{\mathcal{T}_i})d\Pi_{\mathcal{T}}(\mathbf{x}_{\mathcal{T}}) + \tfrac{1}{2}\sigma^2\Delta \int \rho_{t|\mathcal{T}_i}(\mathbf{x}_t|\mathbf{x}_{\mathcal{T}_i})d\Pi_{\mathcal{T}}(\mathbf{x}_{\mathcal{T}}) \tag{76}$$

$$\stackrel{(ii)}{=} -\nabla \cdot (v_t^i(\mathbf{x}_t)\rho_t(\mathbf{x}_t)) + \tfrac{1}{2}\sigma^2\Delta p_t(\mathbf{x}_t), \tag{77}$$

where ($i$) follows from the piece-wise reciprocal property of $\Pi$ for each interval $[t_{i-1}, t_i)$ in (63), and ($ii$) follows by defining:

$$v_t^i(\mathbf{x}_t) = \frac{\int v_{t|\mathcal{T}_i}(\mathbf{x}_t|\mathbf{x}_{\mathcal{T}_i})\rho_{t|\mathcal{T}_i}(\mathbf{x}_t|\mathbf{x}_{\mathcal{T}_i})d\Pi_{\mathcal{T}}(\mathbf{x}_{\mathcal{T}})}{\rho_t(\mathbf{x}_t)} \tag{78}$$

$$= \int v_{t|\mathcal{T}_i}(\mathbf{x}_t|\mathbf{x}_{\mathcal{T}_i})\frac{\rho_{t|\mathcal{T}_i}(\mathbf{x}_t)}{\rho_t(\mathbf{x}_t|\mathbf{x}_{\mathcal{T}_i})}d\Pi_{\mathcal{T}_i}(\mathbf{x}_{\mathcal{T}_i}) \tag{79}$$

$$= \int v_{t|\mathcal{T}_i}(\mathbf{x}_t|\mathbf{x}_{\mathcal{T}_i})d\Pi_{t_i|t}(\mathbf{x}_{t_i}|\mathbf{x}_t) \tag{80}$$

$$= \mathbb{E}_{\Pi_{t_i|t}}\left[\nabla_{\mathbf{x}_t}\log\mathbb{Q}_{t_i|t}(\mathbf{X}_{t_i}|\mathbf{X}_t)|\mathbf{X}_t = \mathbf{x}_t\right]. \tag{81}$$

Hence, for arbitrary $t \in [0, T] - \mathcal{T}$, we get the desired expression:

$$v^\star(t, \mathbf{x}) = \sum_{i=1}^{|\mathcal{T}|}\mathbb{E}_{\Pi_{t_i|t}}v^i(t, \mathbf{x})\mathbf{1}_{[t_{i-1}, t_i)}(t) \tag{82}$$

$$= \sum_{i=1}^{|\mathcal{T}|}\mathbb{E}_{\Pi_{t_i|t}}\left[\nabla_{\mathbf{x}_t}\log\mathbb{Q}_{t_i|t}(\mathbf{X}_{t_i}|\mathbf{X}_t)|\mathbf{X}_t = \mathbf{x}_t\right]\mathbf{1}_{[t_{i-1}, t_i)}(t) \tag{83}$$

Moreover, since the measures induced by the SDEs with drifts $\{v^i\}_{i\in[1:k]}$ for each local interval $\{[t_{i-1}, t_i)\}$ share their marginal distributions at each boundary time points $t \in \mathcal{T}$, the SDEs in (10) form a Markov process. Consequently, we obtain the associated FPE with marginal constraints specified by the prescribed distributions $\{\rho_t\}_{t\in\mathcal{T}}$, which can be constructed using $v^\star$:

$$\partial_t\rho_t = -\nabla \cdot (v_t^\star(\mathbf{x}_t)\rho_t(\mathbf{x}_t)) + \tfrac{1}{2}\Delta\rho_t(\mathbf{x}_t) = 0, \quad \rho_t = \rho_t, \quad \forall t \in \mathcal{T}. \tag{84}$$

This completes the proof. $\qquad\square$

### B.3 Proof of Proposition 3

Our results are based on (Baradat & Léonard, 2020; Mohamed et al., 2021), in a manner analogous to the SBM proof for SBP in (Shi et al., 2024, Appendix C.3), which was based on (Léonard, 2013), and serve as a natural extension to the multi-marginal setting.

**Proposition 3** (Uniqueness). *Let $\mathbb{P}^\star$ be a Markov measure which is reciprocal class of $\mathbb{Q}$ satisfying $\mathbb{P}_t^\star = \rho_t$ for all $t \in \mathcal{T}$. Then, $\mathbb{P}^\star$ is unique solution $\mathbb{P}^{\text{mSBP}}$ of the mSBP.*

*Proof.* Below, we proof that; **(A)**: If some measure $\mathbb{P}^\star \in \mathcal{P}_{[0,T]}$ satisfying the Radon-Nikodym derivative in (43), then it is a Markov process and reciprocal class of $\mathbb{Q}$ satisfying marginal constraints $\{\rho_t\}_{\mathcal{T}}$; **(B)**: If the unique solution $\mathbb{P}^{\text{mSBP}}$ of mSBP exists, then it will has Radon-Nikodym derivative in (43); **(C)**: If some measure $\mathbb{P}^\star \in \mathcal{P}_{[0,T]}$ satisfying the Radon-Nikodym derivative in (43), then it is unique solution $\mathbb{P}^{\text{mSBP}}$ of mSBP.

**(A)** Previously, we established that a solution $\mathbb{P}^\star$ of mSBP possesses a Radon-Nikodym derivative with respect to the reference measure $\mathbb{Q}$ of the product form $\frac{d\mathbb{P}^\star}{d\mathbb{Q}} = \prod_{i=0}^{|\mathcal{T}|}\Psi_{t_i}(\mathbf{x}_{t_i})$ as in (43). By combining the results in (Baradat & Léonard, 2020, Theorem 2.10), since $\prod_{i=0}^{|\mathcal{T}|}\Psi_{t_i}(\mathbf{X}_{t_i})$ is $\sigma(\mathbf{X}_{\mathcal{T}})$-measurable, it is a Markov process. Moreover, it satisfies $\mathbb{P}^\star = \int \mathbb{Q}_{|\mathcal{T}}d\mathbb{P}_{\mathcal{T}}^\star$ in (29), it concluded that $\mathbb{P}^\star$ reciprocal class of $\mathbb{Q}$ *i.e.*, $\mathbb{P}^\star = \mathcal{R}^{\text{mm}}(\mathbb{P}^\star, \mathcal{T})$.

**(B)** Our goal is to verify that $\mathbb{P}^\star$ is indeed the *unique* solution the the mSBP. By combining the results in (Baradat & Léonard, 2020, Theorem 4.5), (if it exists) the unique solution of $\mathbb{P}^{\text{mSBP}}$ is a Markov process and admit following Radon-Nikodym derivative with respect to $\mathbb{Q}$:

$$\frac{d\mathbb{P}^{\text{mSBP}}}{d\mathbb{Q}} = \exp\left(\mathbb{A}[0, 1]\right), \tag{85}$$

where $\mathbb{A}[0, 1]$ is $\sigma(\mathbf{X}_{\mathcal{T}})$-measurable function. Again, since the product form $\prod_{i=0}^{|\mathcal{T}|}\Psi_{t_i}(\mathbf{x}_{t_i})$ is $\sigma(\mathbf{X}_{\mathcal{T}})$-measurable, it states that the solution we found is indeed a unique solution $\mathbb{P}^{\text{mSBP}}$.

**(C)** Let us consider the set of measures satisfying the multi-marginal constraints:

$$\mathcal{C}_{\mathcal{T}} = \{\mathbb{P} \in \mathcal{P}_{[0,T]} : (\mathbf{X}_t)_{\#}\mathbb{P} = \rho_t, \ \forall \ t \in \mathcal{T}\}. \tag{86}$$

By leveraging results in (Mohamed et al., 2021, Theorem 2.6), under mild condition, the unique minimizer $\mathbb{P}^\star \in \mathcal{C}_{\mathcal{T}}$ of mSBP has a Radon-Nikodyn derivative that can be written as $\frac{d\mathbb{P}^\star}{d\mathbb{Q}} = \prod_{i=0}^{|\mathcal{T}|} \Psi_{t_i}(\mathbf{x}_{t_i})$ as in (43).

Combining the arguments from parts **(A-C)**, we establish the following: the unique minimizer $\mathbb{P}^\star \in \mathcal{C}_{\mathcal{T}}$ for mSBP is characterized if and only if it is a Markov process that belongs to the reciprocal class of $\mathbb{Q}$. It concludes the proof.

$\square$

### B.4 Proof of Proposition 4

Our proof builds upon the work of (Peluchetti, 2023; Shi et al., 2024). Standard SBM convergence relies on the Pythagorean property of (reverse) KL-divergence and compactness of the set $\{\mathbb{P} \in \mathcal{P}_{[0,T]} : D_{\mathrm{KL}}(\mathbb{P}|\mathbb{P}^{\mathtt{mSBP}}) \leq D_{\mathrm{KL}}(\mathbb{P}^{(0)}|\mathbb{P}^{\mathtt{mSBP}})\}$. Therefore, if our proposed multi-marginal projection operators, $\mathcal{R}^{\mathtt{mm}}$ and $\mathcal{M}^{\mathtt{mm}}$, also satisfy a Pythagorean law analogous to those in (Peluchetti, 2023; Shi et al., 2024), then their convergence analysis can be directly applied to our multi-marginal scenario.

**Proposition 4** (Convergence). $\mathbb{P}^{(n)} = \mathbb{P}^{mSBP}$ of mSBP as $n \uparrow \infty$ with iterative procedure in (13).

*Proof.* Let $\Pi := \mathcal{R}^{\mathtt{mm}}(\mathbb{P}, \mathcal{T})$ denote the multi-marginal reciprocal projection of a path measure $\mathbb{P}$, and let $\mathbb{M} := \mathcal{M}^{\mathtt{mm}}(\Pi, \mathcal{T})$ be the subsequent multi-marginal Markovian projection of $\Pi$. As established in Proposition 2, the marginal distributions match at each time point, i.e., $\Pi_t = \mathbb{M}_t$ for all $t \in [0, T]$, and specifically at the initial time, $\Pi_0 = \mathbb{M}_0 = \rho_0$. Following the principles outlined by (Peluchetti, 2023, pp. 37-38), we can establish a Pythagorean law for the KL-divergences (Csiszar, 1975) for these multi-marginal projections $\mathcal{R}^{\mathtt{mm}}$ and $\mathcal{M}^{\mathtt{mm}}$. For any path measure $\mathbb{P} \in \mathcal{P}_{[0,T]}$:

$$D_{\mathrm{KL}}(\Pi|\mathbb{P}) = D_{\mathrm{KL}}(\Pi|\mathbb{M}) + D_{\mathrm{KL}}(\mathbb{M}|\mathbb{P}). \tag{87}$$

If we choose $\mathbb{P} = \mathbb{P}^{\mathtt{mSBP}}$, the unique solution to the mSBP, the Pythagorean law implies the inequality:

$$D_{\mathrm{KL}}(\Pi|\mathbb{P}^{\mathtt{mSBP}}) \geq D_{\mathrm{KL}}(\mathbb{M}|\mathbb{P}^{\mathtt{mSBP}}), \tag{88}$$

where equality holds if and only if $\Pi = \mathbb{M}$. Furthermore, as proven in Proposition 3, $\mathbb{P}^{\mathtt{mSBP}}$ is a Markov process within the set of measures $\mathcal{C}_{\mathcal{T}}$ (satisfying the marginal constraints at times $\mathcal{T}$) and belongs to the reciprocal class of the reference measure $\mathbb{Q}$. Consequently, the condition $\Pi = \mathbb{M}$ is met if and only if both $\Pi$ and $\mathbb{M}$ are equal to the unique solution $\mathbb{P}^{\mathtt{mSBP}}$.

Now, consider an iterative process for $n \geq 0$. Let $\mathbb{P}^{(n-1)}$ be a Markovian. Define the reciprocal projection $\Pi^{(n)} = \mathcal{R}^{\mathtt{mm}}(\mathbb{P}^{(n-1)}, \mathcal{T})$. Through the disintegration of the path measure, we have that:

$$D_{\mathrm{KL}}(\Pi^{(n)}|\mathbb{P}^{\mathtt{mSBP}}) = D_{\mathrm{KL}}(\Pi_{\mathcal{T}}^{(n)}|\mathbb{P}_{\mathcal{T}}^{\mathtt{mSBP}}) + \int_{\mathbb{R}^{d \times |\mathcal{T}|}} D_{\mathrm{KL}}(\Pi_{|\mathcal{T}}^{(n)}|\mathbb{P}_{|\mathcal{T}}^{\mathtt{mSBP}})d\Pi_{\mathcal{T}}^{(n)} \tag{89}$$

$$\overset{(i)}{=} D_{\mathrm{KL}}(\Pi_{\mathcal{T}}^{(n)}|\mathbb{P}_{\mathcal{T}}^{\mathtt{mSBP}}), \tag{90}$$

where $(i)$ follows because the reciprocal projection $\mathcal{R}^{\mathtt{mm}}$ ensures that the resulting conditional path measure $\Pi_{|\mathcal{T}}^{(n)}$ is a mixture of bridges between adjacent marginals, identical to that of $\mathbb{P}_{|\mathcal{T}}^{\mathtt{mSBP}}$ (as described in relation to (29)), i.e., $\Pi_{|\mathcal{T}}^{(n)} = \mathbb{P}_{|\mathcal{T}}^{\mathtt{mSBP}} = \mathbb{Q}_{|\mathcal{T}}$. Next, let $\mathbb{M}^{(n)} = \mathcal{M}^{\mathtt{mm}}(\Pi^{(n)}, \mathcal{T})$ be the Markovian projection of $\Pi^{(n)}$. The KL divergences between $\mathbb{M}^{(n)}$ and $\mathbb{P}^{\mathtt{mSBP}}$ becomes:

$$D_{\mathrm{KL}}(\mathbb{M}^{(n)}|\mathbb{P}^{\mathtt{mSBP}}) = D_{\mathrm{KL}}(\mathbb{M}_{\mathcal{T}}^{(n)}|\mathbb{P}_{\mathcal{T}}^{\mathtt{mSBP}}) + \int_{\mathbb{R}^{d \times |\mathcal{T}|}} D_{\mathrm{KL}}(\mathbb{M}_{|\mathcal{T}}^{(n)}|\mathbb{P}_{|\mathcal{T}}^{\mathtt{mSBP}})d\mathbb{M}_{\mathcal{T}}^{(n)} \tag{91}$$

$$\overset{(ii)}{\geq} D_{\mathrm{KL}}(\Pi_{\mathcal{T}}^{(n+1)}|\mathbb{P}_{\mathcal{T}}^{\mathtt{mSBP}}), \tag{92}$$

where $\Pi^{(n+1)} = \mathcal{R}^{\mathtt{mm}}(\mathbb{M}^{(n)}, \mathcal{T})$, and $(ii)$ is stated to follow from the IMF iteration as per (24). This implies the desired intermediate result for the convergence argument:

$$D_{\mathrm{KL}}(\mathbb{M}^{(n)}|\mathbb{P}^{\mathtt{mSBP}}) \geq D_{\mathrm{KL}}(\Pi^{(n+1)}|\mathbb{P}^{\mathtt{mSBP}}). \tag{93}$$

Consequently, under the assumption that the relevant KL divergences (such as $D_{\mathrm{KL}}(\Pi|\mathbb{M})$) are finite, the convergence proof presented by (Shi et al., 2024, Appendix C.6) can be directly adapted to our multi-marginal case, given our construction of $\mathcal{M}^{\mathtt{mm}}$ and $\mathcal{R}^{\mathtt{mm}}$. This concludes the proof. $\square$

### B.5 Proof of Corollary 5

**Corollary 5** (Multi-Marginal Schrödinger Bridge). *Let us assume a sequence of controls $\{v^i, u^i\}_{i \in [1:k]}$, where each $v^i, u^i$ induced local SBs $\mathbb{P}^i$ of SBP over local interval $[t_{i-1}, t_i]$ with distributions $(\rho_{t_{i-1}}, \rho_{t_i})$ in a forward and backward direction, respectively. If $\lim_{t \uparrow t_i} v^i(t, \mathbf{x}) = v^{i+1}(t, \mathbf{x})$ and $\lim_{t \downarrow t_{i-1}} u^i(t, \mathbf{x}) = u^{i-1}(t, \mathbf{x})$ for all $i \in [1:k]$, then $\mathbb{P}^{mSBP}$ of mSBP induced by following SDEs:*

$$d\mathbf{X}_t^\star = [f_t(\mathbf{X}_t^\star) + \sigma v^\star(t, \mathbf{X}_t^\star)]\, dt + \sigma d\mathbf{W}_t, \quad \mathbf{X}_0^\star \sim \rho_0. \tag{18a}$$

$$d\mathbf{Y}_t^\star = [-f_{T-t}(\mathbf{Y}_t^\star) + \sigma u^\star(t, \mathbf{Y}_t^\star)]\, dt + \sigma d\mathbf{W}_t, \quad \mathbf{Y}_0^\star \sim \rho_T, \tag{18b}$$

$$where \quad v^\star(t, \mathbf{x}) = \sum_{i=1}^k \mathbf{1}_{[t_{i-1}, t_i)}(t) v^i(t, \mathbf{x}), \quad u^\star(t, \mathbf{x}) = \sum_{i=1}^k \mathbf{1}_{(t_{i-1}, t_i]}(t) u^i(t, \mathbf{x}). \tag{18c}$$

*Proof.* Consider local forward SBs $\overrightarrow{\mathbb{P}}^i$ (governed by control $v^i$) and local backward SBs $\overleftarrow{\mathbb{P}}^i$ (governed by $u^i$) on intervals $[t_{i-1}, t_i]$. Global forward path measure $\overrightarrow{\mathbb{P}} = \rho_0 \prod_{i=1}^k \overrightarrow{\mathbb{P}}^i_{|t_{i-1}}$ and global backward path measure $\overleftarrow{\mathbb{P}} = \rho_T \prod_{i=1}^k \overleftarrow{\mathbb{P}}^i_{|t_i}$ are constructed by sequentially composing these local SBs, starting from the initial distribution $\rho_0$ and terminal distribution $\rho_T$, respectively. By this construction, $\overrightarrow{\mathbb{P}}$ and $\overleftarrow{\mathbb{P}}$ inherently satisfy all specified marginal constraints $\{\rho_t\}_{t \in \mathcal{T}}$ and belong to the reciprocal class of the reference measure $\mathbb{Q}$. The Markov property and absolute continuity constraint $D_{\mathrm{KL}}(\overrightarrow{\mathbb{P}}|\mathbb{Q}) < \infty$ or $D_{\mathrm{KL}}(\overleftarrow{\mathbb{P}}|\mathbb{Q}) < \infty$ for these global path measures $\overrightarrow{\mathbb{P}}$ and $\overleftarrow{\mathbb{P}}$ hinges on the continuity of their sample paths $\mathbf{X}_t^\star$. This path continuity is achieved if the composite global controls $v^\star$ in (18a) and $u^\star$ in (18b) are continuous across the entire time horizon $[0, T]$, ensuring seamless transitions at each intermediate time $t_i$ *i.e.*, $\lim_{t \uparrow t_i} v^i(t, \mathbf{x}) = v^{i+1}(t, \mathbf{x})$ and $\lim_{t \downarrow t_{i-1}} u^i(t, \mathbf{x}) = u^{i-1}(t, \mathbf{x})$ for all $i \in [1:k]$. With continuous sample paths, $\overrightarrow{\mathbb{P}}$ and $\overleftarrow{\mathbb{P}}$ are indeed Markov processes satisfying the absolute continuity condition with respect to $\mathbb{Q}$. Given Proposition 3, which states that a Markov process satisfying all marginal constraints and belonging to the reciprocal class of $\mathbb{Q}$ is the unique solution to mSBP. It concludes the proof. $\square$

## C  Experimental Details

Our evaluation of MSBM involved several datasets and followed established experimental protocols from baseline methods to ensure fair comparisons. For the petal dataset and the 100-dimensional EB (Moon et al., 2019), we adopted the experimental setup of DMSB (Chen et al., 2023)[3]. The processed data for these experiments were inherited from TrajectoryNet (Tong et al., 2020)[4]. To maintain a fair comparison with DMSB, we parameterized both the forward ($v$) and backward ($u$) controls using the residual-based networks described in (Chen et al., 2023), ensuring a similar total model parameter count of approximately 1.28M for these two datasets. For the 100-dimensional EB dataset, we further split the entire dataset into training and testing subsets with an 85%/15% ratio, respectively.

Regarding the hESC (Chu et al., 2016), our experiments mirrored the setup of SBIRR (Shen et al., 2024)[5] and inherited the processed data therein. We based the parameterization of our model on the network architecture used for the petal and 100-dim EB datasets. For a consistent comparison on the hESC dataset, we maintained a model size of approximately 24k total parameters.

---

[3] https://github.com/TianrongChen/DMSB, under MIT license
[4] https://github.com/KrishnaswamyLab/TrajectoryNet, Non-Commercial License Yale Copyright
[5] https://github.com/YunyiShen/SB-Iterative-Reference-Refinement

Table 5: Training Hyper-parameters

| Dataset | Learning Rate | Iteration (N) | Training step (S) | # of Discretization | Batch Size | $T$ |
|---|---|---|---|---|---|---|
| Petal | $1 \times 10^{-3}$ | 20 | 1000 | 30 | 256 | 4 |
| hESC | $1 \times 10^{-3}$ | 100 | 1000 | 30 | 256 | 5 |
| EB(100-dim) | $2 \times 10^{-4}$ | 10 | 1000 | 100 | 256 | 4 |
| EB(5-dim) | $2 \times 10^{-4}$ | 3 | 50000 | 100 | 256 | 4 |
| CITE(5-dim) | $1 \times 10^{-4}$ | 50 | 2000 | 100 | 256 | 3 |
| CITE(100-dim) | $1 \times 10^{-4}$ | 50 | 2000 | 100 | 256 | 3 |
| MUILTI(5-dim) | $1 \times 10^{-4}$ | 50 | 2000 | 100 | 256 | 3 |
| MULTI(5-dim) | $1 \times 10^{-4}$ | 50 | 2000 | 100 | 256 | 3 |

For the 5-dimensional EB dataset, we followed the experimental protocol of NLSB (Koshizuka & Sato, 2022)[6]. We inherited processed data from TrajectoryNet (Tong et al., 2020). In this case, the forward ($v$) and backward ($u$) controls were also parameterized with the residual-based networks described in (Chen et al., 2023).

For the CITE and MULTI datasets, we followed the experimental protocol of OT-CFM and the forward ($v$) and backward ($u$) controls are parameterized in the same way as in 5-dim EB case.

Across all experiments, models were trained using the Adam optimizer (Kingma, 2014) with Exponential Moving Average (EMA) applied at a decay rate of 0.999. The proposed MSBM training procedure (detailed in Algorithm 1) involved $N$ outer iterations, with each outer iteration containing $S$ inner training steps. Cached marginal distributions were updated in each outer iteration. Impressively, the complete MSBM training for all datasets was accomplished in less than one hour using a single NVIDIA A6000 GPU excetp CITE and MULTI datasets, where we use a single NVIDIA RTX 3090. The remaining training hyper-parameters are detailed in Table 5.

Additionally, the tables below present the complete results from experiments conducted three times, each using a different random seed.

Table 6: Full results over 3 different seeds. Performance on the 100-dim PCA of EB dataset. MMD and SWD are computed between test $\rho_{t_i}^{\texttt{te}}$ and generated $\hat{\rho}_{t_i}$ by simulating the dynamics from test $\rho_{t_0}^{\texttt{te}}$.

| Methods | MMD ↓ | | | | SWD ↓ | | | |
|---|---|---|---|---|---|---|---|---|
| | Full | $t_1$ | $t_2$ | $t_3$ | Full | $t_1$ | $t_2$ | $t_3$ |
| NLSB[†] (Koshizuka & Sato, 2022) | 0.66 | 0.38 | 0.37 | 0.37 | 0.54 | 0.55 | 0.54 | 0.55 |
| MIOFlow[†] (Huguet et al., 2022) | 0.23 | 0.23 | 0.90 | 0.23 | 0.35 | 0.49 | 0.72 | 0.50 |
| DMSB[†] (Chen et al., 2023) | 0.03 | **0.04** | **0.04** | **0.04** | 0.16 | 0.20 | 0.19 | **0.18** |
| **MSBM** | **0.018**± 4e-4 | **0.049**± 3e-2 | **0.038**± 5e-4 | 0.05± 9e-4 | **0.129**± 3e-3 | **0.1895**± 1e-2 | **0.1772**± 2e-3 | 0.1997± 4e-3 |

† result from (Chen et al., 2023).

Table 7: Full results over 3 different seeds. Performance on the 5-dim PCA of hESC dataset. $\mathcal{W}_2$ is compute between test $\rho_{t_i}$ and generated $\hat{\rho}_{t_i}$ by simulating the dynamics from test $\rho_{t_0}$.

| Methods | $\mathcal{W}_2$ ↓ | | Runtime |
|---|---|---|---|
| | $t_1$ | $t_3$ | hours |
| TrajectoryNet[†] | 1.30 | 1.93 | 10.19 |
| DMSB[†] | 1.10 | 1.51 | 15.54 |
| SBIRR[†] | **1.08** | 1.33 | 0.36 (0.38)* |
| **MSBM** (Ours) | 1.083 ± 7e-3 | **1.304** ± 3e-2 | **0.09** ± 1e-2 |

† result from (Shen et al., 2024).

[6]https://github.com/take-koshizuka/NLSB, under MIT license

Table 8: Full results over 3 different seeds. Performance on the 5-dim PCA of EB dataset. $\mathcal{W}_1$ is computed between test $\rho_{t_i}^{\text{te}}$ and generated $\hat{\rho}_{t_i}$ by simulating the dynamics from previous test $\rho_{t_{i-1}}^{\text{te}}$.

| Methods | $\mathcal{W}_1 \downarrow$ | | | | |
| --- | --- | --- | --- | --- | --- |
| | $t_1$ | $t_2$ | $t_3$ | $t_4$ | Mean |
| Neural SDE[†] (Li et al., 2020) | 0.69 | 0.91 | 0.85 | 0.81 | 0.82 |
| TrajectoryNet[†] (Tong et al., 2020) | 0.73 | 1.06 | 0.90 | 1.01 | 0.93 |
| IPF (GP)[†] (Vargas et al., 2021) | 0.70 | 1.04 | 0.94 | 0.98 | 0.92 |
| IPF (NN)[†] (Bortoli et al., 2021) | 0.73 | 0.89 | 0.84 | 0.83 | 0.82 |
| SB-FBSDE[†] (Chen et al., 2022a) | **0.56** | 0.80 | 1.00 | 1.00 | 0.84 |
| NLSB[†] (Koshizuka & Sato, 2022) | 0.68 | 0.84 | 0.81 | 0.79 | 0.78 |
| OT-CFM[†] (Tong et al., 2024) | 0.78 | 0.76 | 0.77 | 0.75 | 0.77 |
| WLF-SB[‡] (Neklyudov et al., 2023b) | 0.63 | 0.79 | 0.77 | **0.75** | 0.73 |
| MSBM (Ours) | 0.64± 7e-3 | **0.73**± 8e-3 | **0.72**± 1e-2 | **0.73**± 9e-3 | **0.71**± 7e-3 |

† result from (Koshizuka & Sato, 2022), ‡ result from (Neklyudov et al., 2023b).

