# OpenReview forum: "Multi-Marginal Schrödinger Bridge Matching"
_TMLR — Under review for TMLR_

### Review · Reviewer_WTpj · 2026-04-17

**Summary Of Contributions:**

Given distributions of several time points, identifying its dynamics, that is, interpolated distributions is useful to understand and artificially generate the data.

Suppose a continuous-time time-series where each timepoint is represented as a measure (probabilistic distribution), which is called the *path measure*. Then, the *Schrödinger Bridge* problem is defined as finding the path measure $\\mathbb{P}$ under the following conditions:
- Suppose certain time dependency (e.g., Brownian motion) in $\\mathbb{P}$.
- $\\mathbb{P}\_0$ and $\\mathbb{P}\_T$ (measures of the beginning and the ending timepoints) are constrained.
- Under the conditions above, given a reference path measure $\\mathbb{Q}$, find $\\mathbb{P}$ which is most similar to $\\mathbb{Q}$ in the sense of Kullback-Leibler divergence.

Proposed method extends the Schrödinger Bridge problem to the case when measures at any (finite) number of timepoints are constrained. They extended existing algorithm *Schrödinger Bridge Matching*, called *Multi-Marginal Schrödinger Bridge Matching*, for this setup. The basic strategy is to apply Schrödinger Bridge Matching for each segment of timepoints, but just doing so does not work well: how to concatenate them, how to avoid error accumulations, and how to parallelize are main issues to be solved.

**Audience:**

Yes

**Audience Explanation:**

The method handles "time-varying distributions", and interpolating them is useful to understand such data.

**Broader Impact Concerns:**

No particular concern.

**Claims And Evidence:**

No

**Claims Explanation:**

Due to mathematical defects, it was difficult to follow the method.

**Requested Changes:**

## Points critical to securing my recommendation for acceptance

- Overall: I felt that there are many issues in mathematical notations. Please see "Notation issues" below.
- Overall: It was difficult for me to first imagine why we would like to interpolate measures (distributions) defined for timepoints rather than single-valued vectors. Although several examples are presented, the reason to use measures was still unclear. Perhaps, explaining the inputs and outputs of the problem we solve using a similar figure to Figure 2 may be helpful.
- Section 3: It states that the dynamic multi-marginal Schrödinger Bridge problem itself is formulated in (Chen et al., 2019), but what is the advantage of solving it by SBM-based algorithm compared to the algorithm presented in (Chen et al., 2019)?
- Section 3.2: The advantage of approximating $v^*$ via a neural network is unclear.

### Notation issues

- Section 1, "Notation": It first states that "Let $P\_{[0, T]}$ denote the space of continuous functions taking values in $\\mathbb{R}^d$ on the interval $[0, T]$", but then states that "We use an uppercase letter $\\mathbb{P}\\in P[0, T]$ to represent a path measure". However, as far as reading solely the first text, it looks that each element of $P\_{[0, T]}$ is a mere function (that returns single real number for any $t\\in[0, T]$), not a path measure (i.e., returns a distribution for any $t\\in[0, T]$). Please explicitly define $P\_{[0, T]}$ to represent the latter.
- Section 1, "Notation": Perhaps it is because I am not familiar with SDE, I could not understand the definition of "the conditional distribution of $\\mathbb{P}$ given \\mathcal{T}". Although it assumes the Brownian motion, why conditioning only by time points (without the values at the time points) can compute the conditional distributions?
- Section 2.1: What are $\\mathbf{X}\_t$ and $\\mathbf{X}\_0$? Just meaning "elements in a path measure $(\\mathbf{X}\_t)\_{t\\in[0, T]}$"? Please explicitly define.
- Section 2.1: Although it states that $\\mathbf{W}\_t$ is a Brownian motion, please provide specific definition of $\\mathbf{W}\_t$ (if several defenitions are available, please state the condition that $\\mathbf{W}\_t$ must satisfy).
- Section 2.1, "Reciprocal Projection": How $\\mathbb{P}\_\\mathcal{T} \\mathbb{Q}\_{|\\mathcal{T}}$ (product of two path measures) is defined? Just meaning the elementwise product (i.e., $\[\\mathbb{P}\_\\mathcal{T} \\mathbb{Q}\_{|\\mathcal{T}}\](t) := \\mathbb{P}\_\\mathcal{T}(t) \\mathbb{Q}\_{|\\mathcal{T}}(t)$ for any $t\\in[0, T]$)? Please explicitly define.
- Section 3.1: In Proposition 1, the condition that $\\mathbb{Q}$ must satisfy is not given in the main text, although in the proof (Appendix B.1) it assumes that $\\mathbb{Q}$ is a Markov measure. Please write also in the main text that $\\mathbb{Q}$ is assumed to be a Markov measure.

## Points that simply strengthen the work in my view

- Section 3.2: The procedure to approximate $v^*$ is achieved by a similar computation to the training, but I felt that it is not good to be called as "training" but as just "minimization".

## Minor fixes

- Overall: Please refer the section numbers of appendices at appropriate places in the main text.
- Abstract: "While Schr\\"{o}dinger Bridge (SB) offer" -> "While Schr\\"{o}dinger Bridge (SB) offers"
- Section 1: "algoritmh" -> "algorithm"
- Section 6 (before Section 6.1): Stating the name "petal dataset" before showing the source (Section 6.1) is confusing. Please list up all datasets in Section 6 (before 6.1).

---

### Review · Reviewer_1BZx · 2026-05-25

**Summary Of Contributions:**

The paper extends the iterative Markovian fitting (IMF) procedure for solving the Schrödinger bridge problem to multi-marginal cases, where instead of just matching the endpoint distributions, we have a sequence of marginals $p(x_1), p(x_2), …, p(x_T)$. The work is a part of a larger literature on extending Schrödinger bridge solving methods to multi-marginal cases. The key mathematical and algorithmic extensions are noticing that the loss function and training process factorises into a separate Markovian fitting problem for each interval, sidestepping the need for simulating SDEs across the entire [0,T] interval and thus avoiding compounding approximation errors and long sequential computations, and 2) using a shared network for the intervals to guarantee continuity across [0,T]. The method performs better than the previous methods on most datasets tried, and often is much faster to train.

**Audience:**

Yes

**Audience Explanation:**

There is clearly a sub-literature in multi-marginal Schrödinger bridge solvers, and the paper is a legitimate addition and improvement in this line of work.

**Broader Impact Concerns:**

I do not have any ethical concerns regarding the paper.

**Claims And Evidence:**

Yes

**Claims Explanation:**

The mathematical claims include proofs in the Appendix, and the claims regarding improvements over baselines are supported by the experimental evidence.

**Requested Changes:**

The paper contained a few typos:

“iteraive procedure” -> “iterative procedure”

“we extends the” -> “we extend the”

“the uniquness condition” -> “the uniqueness condition”

“P⋆ is unique solution” -> “is the unique solution”